# On the Universal Statistical Consistency of Expansive Hyperbolic Deep Convolutional Neural Networks

## Abstract

The emergence of Deep Convolutional Neural Networks (DCNNs) has been a pervasive tool for accomplishing widespread applications in computer vision. Despite its potential capability to capture intricate patterns inside the data, the underlying embedding space remains Euclidean and primarily pursues contractive convolution. Several instances can serve as a precedent for the exacerbating performance of DCNNs. The recent advancement of neural networks in the hyperbolic spaces gained traction, incentivizing the development of convolutional deep neural networks in the hyperbolic space. In this work, we propose Hyperbolic DCNN based on the Poincaré Ball. The work predominantly revolves around analyzing the nature of expansive convolution in the context of the non-Euclidean domain. We further offer extensive theoretical insights about the universal consistency of the expansive convolution in the hyperbolic space. Several simulations were performed not only on the synthetic datasets but also on some real-world datasets. The experimental results reveal that the hyperbolic convolutional architecture outperforms the Euclidean ones by a commendable margin.

## 1 Introduction

The ubiquitous utility of Deep Convolutional Neural Networks (DCNNs) LeCun et al. (1998) dominated the arena of Computer Vision Yang & Li (2017); Fu et al. (2019); Sonata et al. (2021) over the past decade. This profound success can be attributed to the effectiveness of the CNNs in approximating the broader class of continuous functions Lin et al. (2022b). The prevalent convolutional neural architectures He et al. (2016); Simonyan (2014) predominantly operate in the Euclidean feature space. The choice of Euclidean space is mostly for implementable closed-form vector space and inner product structures, and their availability in tabular forms. We are focused on DCNN architectures which evolve around $1-$ dimensional convolution based on one input channel and ReLU (Rectified Linear Unit, $r(x) := \max(0, x)$ for $x \in \mathbb{R}$) activation function given to the computational units (Neurons). For two functions $f, g : \mathbb{R}^n \to \mathbb{R}$, we define their convolution as

$$f \otimes g(z) := \int_{\mathbb{R}^n} f(x)g(z-x)dx,$$

where $z \in \mathbb{R}^n$. In the discrete version, given a filter $\mathbf{w} := \{\mathbf{w_i}\}_{\mathbf{i}=-\infty}^{\infty}$, where only finitely many $w_j \neq 0$. We call $\mathbf{w}$ to be a filter of length $s$ if $w_j \neq 0$ only for $0 \leq j \leq s$. For a one dimensional input vector $\mathbf{v} := \{\mathbf{v_1}, \mathbf{v_2}, ..., \mathbf{v_n}\} \in \mathbb{R}^{\mathbf{n}}$, we can define two types of convolution operations, namely *Expansive Convolution* $(\mathbf{w} * \mathbf{v})$ and *Contractive Convolution* $(\mathbf{w} \star \mathbf{v})$, given by the following forms of equations

$$(\mathbf{w} * \mathbf{v})_k := \sum_{i=1}^{n} w_{k-i}v_i, \ \ k = 1, 2, ..., n+s \tag{1}$$

and

$$(\mathbf{w} \star \mathbf{v})_k := \sum_{i=k-s}^{k} w_{k-i}v_i, \ \ k = s+1, s+2, ..., n \tag{2}$$

respectively. Now for a set of $L$ filters $\{\mathbf{w_i}\}_{\mathbf{i=1}}^{\mathbf{L}}$, where $L$ is the depth of our network with $L$ many bias vectors $\{b_i\}_{i=1}^{L}$, we recursively define the output of an intermediate layer given in terms of the output of the previous layer as Zhou (2020a)

$$h_i(x) = r\left(\mathbf{w}_i \circ h_{i-1}(x) + b_i\right), \text{ for } i = 1, 2, ..., L,$$

starting with the input as $h_0(x) = x$ and $\circ$ can be either $*$ or $\star$ as defined by equations 1 or 2 respectively. The final one-dimensional output of this network is defined as the scalar product between the output produced by the $L^{th}$ layer with a trainable vector $a_L$ of compatible length

$$h_o(x) := a_l \cdot h_L(x).$$

Although this form of Euclidean convolution has been proven to be enormously successful in several tasks in computer vision, several precedents Lin et al. (2022a); Djeddal et al. (2021); Long & van Noord (2023) can be put forth where Euclidean feature space seems unproductive, like datasets containing hierarchical structures. Learning embeddings of hierarchical data in Euclidean spaces often falls short in capturing meaningful structural information. To address this limitation, researchers have explored neural network architectures in non-Euclidean spaces, particularly hyperbolic geometry Ganea et al. (2018); Nickel & Kiela (2017b); Bdeir et al. (2023). Hyperbolic Neural Networks (HNNs) Ganea et al. (2018) have emerged as a promising framework, leveraging negatively curved spaces to better represent complex relationships and hierarchical structures. Building on this, Hyperbolic Deep Convolutional Neural Networks (HDCNNs) have shown success in various image-related tasks. However, despite these empirical advances, a rigorous theoretical understanding of hyperbolic convolution remains largely undeveloped.

Motivated by this gap, our work provides a comprehensive statistical analysis of hyperbolic convolution, focusing on the consistency of expansive convolutional operations in hyperbolic space. This topic has received little attention compared to its Euclidean counterpart. Prior studies, such as Lin et al. (2022b), have established consistent results for Euclidean convolutional networks using bounds on packing numbers and error analysis. Yet, similar theoretical foundations for hyperbolic networks are lacking.

To this end, we introduce a theoretical framework for 1-D expansive Hyperbolic Deep Convolutional Neural Networks (eHDCNNs), extending the conventional Euclidean DCNNs to the hyperbolic domain via the Poincaré Ball model. This foundation enables the formulation of key statistical properties and paves the way for theoretical consistency analysis. Empirical results on both synthetic and real-world datasets confirm the superiority of hyperbolic representations, with significantly faster convergence and lower error rates compared to Euclidean models, thus validating our theoretical contributions.

**Contribution**

Our main contributions could be summarized in the following way:

- We provide theoretical insights, including the consistency analysis of the expansive 1-D convolution in hyperbolic space. To the best of our knowledge, this is the first work to present a complete proof in the context of a fully hyperbolic set-up. In doing so, we have also introduced the concept of a fully hyperbolic convolution operation on the Poincaré Ball, which is the generalization of the conventional Euclidean convolution operation on hyperbolic spaces. Additionally, we have extended several well-known statistical terminologies like population risk, empirical risk minimizer, and regression estimator in the ambit of the hyperbolic framework to derive universal consistency. All necessary proofs and derivations are provided in Section A.

- Our experimental simulations demonstrate that eHDCNN training converges more rapidly than the training of the Euclidean DCNN, which we have already established theoretically. The faster reduction of error rate reaffirms the requirements of the lower number of training iterations for hyperbolic convolutional networks compared to their conventional Euclidean counterparts, establishing the effectiveness of eHDCNN. Details of the experiments and simulations are provided in Section 7.

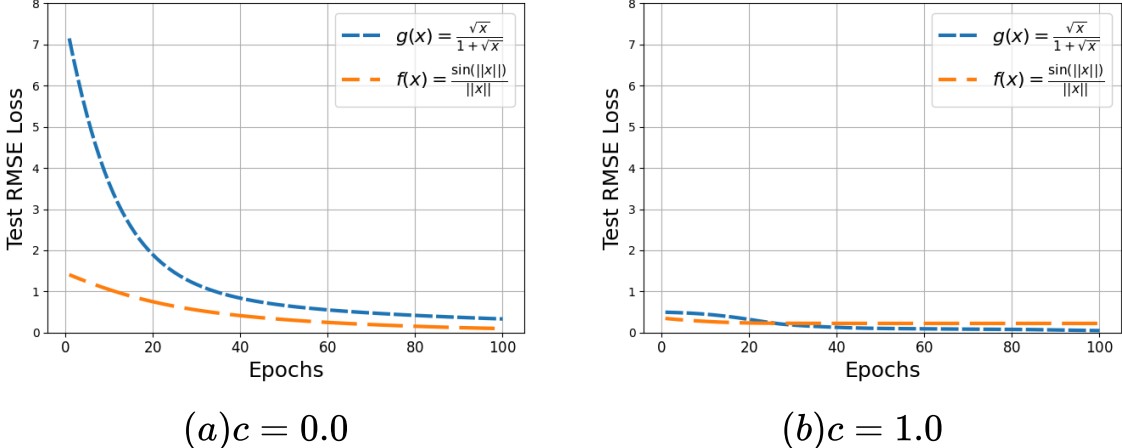

Figure 1: Test Root Mean Squared error for $f(x)$ and $g(x)$ plotted using (a) eDCNN architecture curvature 0 (i.e., Euclidean space) and (b) eHDCNN architecture with curvature 1.

## 2 A Motivating Example

We will demonstrate the efficacy of our proposed hyperbolic expansive convolution over conventional Euclidean expansive convolution through the following simulation.

**Experimental Setup**

Consider the following functions,

$$f(x) = \frac{\sin(\|x\|_2)}{\|x\|_2}, \quad g(x) = \frac{\sqrt{\|x\|_2}}{1 + \sqrt{\|x\|_2}},$$

where $f(x)$ and $g(x)$ both are modeled as regression task like $y := h(x) + \epsilon$. Here, $h$ can be replaced with either $f$ or $g$. The training instances are generated by sampling $\epsilon \sim \mathcal{N}(0, 0.01)$ and $x \sim unif([-1, 1]^5)$. A total of 1000 instances will be generated for both cases, where 800 samples will be used for training and 200 samples will be used for testing. Importantly, the test samples are considered without the Gaussian noise. The filter length is fixed at 8 with the number of layers being 4. Both models are trained for 100 epochs over the training set. The test Root Mean Squared Loss (RMSE) is recorded after the completion of training and presented in Figure 1. Experiments are conducted for two different curvatures, $c = 0$ (Euclidean space) and $c = 1$. We considered unit radius Poincaré Ball as the hyperbolic space. Assuming the point set is in a discrete metric space, we employed Gromov Hyperbolicity (GH) Väisälä (2005) to measure the hyperbolicity ($\delta$) of the corresponding data points. The metric offers hyperbolicity of $f$ and $g$ are respectively $\delta_f = 0.45$ and $\delta_g = 0.017$, indicating that $g$ is more hyperbolic comparing to $f$, which is also supported by our experimental observation, since the Test RMSE curve for $g(x)$ being lower compared to $f(x)$ in curvature 1.0 at higher epochs.

The choices of the functions $f$ and $g$ are not arbitrary. Rather, they are motivated by the observation that the datasets they generate exhibit low Gromov Hyperbolicity (GH) indices, with one of the values close to 0 and less than 1 in both cases. This suggests that the underlying geometry of the data is highly hierarchical or tree-like. Such structures are known to benefit from hyperbolic representations, where models like eHDCNN can more naturally and efficiently capture these relationships Sala et al. (2018b); Yim & Gilbert (2023). Lower GH indices correspond to greater hyperbolicity, reinforcing the need for hyperbolic architectures. Our empirical observation further supports this, that the dataset generated by $g$ achieves lower test RMSE at higher negative curvature values after sufficient training, aligning with the theoretical suitability of hyperbolic models for highly hyperbolic data.

## 3 Related Works

**Hyperbolic Image Embedding and NLP Tasks**

Developing a Hyperbolic Neural network for computer vision tasks has been mainly focused on combining Euclidean Encoders and Hyperbolic Embedding. These architectures were demonstrated to be effective in performing various vision tasks, for example, recognition Khrulkov et al. (2020),Guo et al. (2022a), generation Nagano et al. (2019), and image segmentation Atigh et al. (2022). While Hyperbolic Embedding has also been tremendously successful in performing various tasks related to Natural Language Processing Nickel & Kiela (2017a),Nickel & Kiela (2018). These ideas were mainly motivated by the expressive power of the hyperbolic spaces to represent graph or tree-like hierarchies in shallow dimensions with very low distortions. However, deploying Riemannian Optimization algorithms to train this architecture is difficult due to the inability to extend them for visual data since NLP tasks lack the availability of discrete data Sala et al. (2018a),Sarkar (2011).

**Fully Connected Hyperbolic Neural Network**

In 2018 Ganea et al. (2018), and in 2020 Shimizu et al. (2020) independently developed the structure of Hyperbolic Neural Networks on Poincaré Ball by utilizing the gyrovector space structure. They defined the generalized notions of different layers like fully connected, convolutional, or attention layers. Fan et al. (2022), Qu & Zou (2022) tried to develop variations of HNN models like fully Hyperbolic GAN on Lorentz Model space, van Spengler et al. (2023) proposed a fully hyperbolic CNN architecture on Poincaré Ball model. Very recently, Bdeir et al. (2023) presented a fully convolutional neural network on the Lorentz Model to perform complex computer vision tasks, where they generalized fundamental components of CNNs and proposed novel formulations of convolutional layer, batch normalization, and Multinomial Logistic Regression (MLR) classifier. Moreover, hyperbolic graph neural networks can also accomplish recommendation tasks. There are numerous recommender systems such as graph neural collaborative filtering Sun et al. (2021), Yang et al. (2022), social network enhanced network system Wang et al. (2021), knowledge graph enhanced recommender system Chen et al. (2022), and session-based recommender system Guo et al. (2022a), Li et al. (2021).

**Batch Normalization in Hyperbolic Neural Networks**

Batch Normalization Ioffe (2015) restricts the internal departure of neuron outputs by normalizing the outputs produced by the activations at each layer. This adds stability to the training procedure and speeds up the training phase. Several attempts have been made to transcend the normalization of conventional neural networks in the hyperbolic setup. The general framework of Riemannian Batch Normalization Lou et al. (2020), however, suffers from slower computation and iterative update of the Frechét centroid, which does not arise from Gyrovector Group properties. Additionally, Bdeir et al. (2023) proposed an efficient batch normalization algorithm based on the Lorentz model, utilizing the Lorentz centroid and a mathematical rescaling operation.

**Numerical Stability of Hyperbolic Neural Networks**

Training of Hyperbolic Neural Networks developed on the Lorentz Model can lead to instability and floating point error due to rounding since the volume of the Lorentz model grows exponentially with respect to radius. Sometimes, people work with these floating-point representations in 64-bit precision with a higher memory cost. Mishne et al. (2023),Guo et al. (2022b),Mathieu et al. (2019) proposed some versions of feature clipping and Euclidean reparameterization to mitigate these issues. However, they largely overlooked some critical aspects, such as defining a fully hyperbolic convolutional layer or classifiers like MLR, which are essential for various computer vision tasks. In this paper, we fully address this gap by developing a novel architecture from the ground up, along with the theory of its universal consistency.

## 4 Preliminaries

This section discusses the preliminaries of Riemannian Manifolds and Hyperbolic Geometry, which are beneficial for understanding our proposed framework.

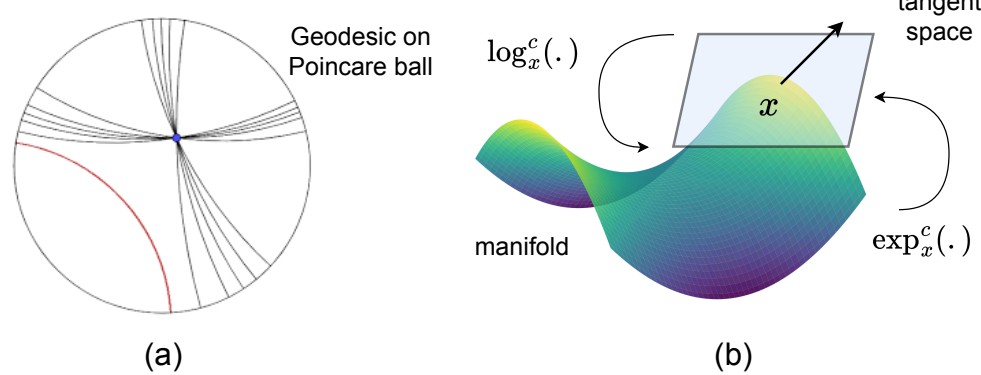

Figure 2: (a) The geodesics in Poincaré ball and (b) Riemannian manifolds with Tangent space are presented. The functionalities of the logarithmic map and exponential map are also illustrated.

## 4.1 Riemannian Manifold, Tangent Space, and Geodesics

An $n$-dimensional *manifold* $\mathcal{M}$ is a geometric space that, in small neighborhoods, behaves like the familiar Euclidean space $\mathbb{R}^n$ Tu (2017). At each point $x \in \mathcal{M}$, we can define a corresponding *tangent space* $T_x(\mathcal{M})$, which captures the possible directions in which one can move from $x$—similar to a flat plane touching a curved surface.

If each of these tangent spaces is equipped with an inner product (i.e., a way to measure angles and lengths), then $\mathcal{M}$ becomes a *Riemannian manifold* do Carmo (1992). This inner product is given by a family of functions $g = \{g_x : T_x(\mathcal{M}) \times T_x(\mathcal{M}) \to \mathbb{R}\}$, one for each point $x \in \mathcal{M}$. Using this, we can define the length of a curve $\gamma : [a, b] \to \mathcal{M}$ as:

$$L(\gamma) := \int_a^b \sqrt{g_{\gamma(t)}(\gamma'(t), \gamma'(t))} \, dt,$$

which in turn gives a way to define distances between points on the manifold. The shortest such curve between two points is called a *geodesic*, and the distance along this curve is the *geodesic distance*.

Moreover, Riemannian manifolds allow us to define curvature. For two linearly independent directions $u$ and $v$ at a point $x$, the *sectional curvature* is defined as:

$$k_x(u, v) := \frac{g_x(R(u, v)v, u)}{g_x(u, u)g_x(v, v) - g_x(u, v)^2},$$

where $R$ is the Riemann curvature tensor and $\nabla$ is the Riemannian connection—used to describe how vectors change along curves.

## 4.2 Poincaré Ball Model

A *hyperbolic space* is a Riemannian manifold where the sectional curvature is negative and constant throughout the space. Among various representations of hyperbolic space, we use the *Poincaré Ball Model* due to its elegant geometry and convenience Lee (2006). For a given curvature $k < 0$ (denoted as $-c$), the $n$-dimensional Poincaré Ball $\mathbb{D}^n$ is defined as the open ball of radius $1/\sqrt{c}$ centered at the origin in $\mathbb{R}^n$.

In this model, geodesics are represented by arcs of circles that intersect the boundary of the ball at right angles. The geodesic distance between two points $p, q \in \mathbb{D}^n$ is given by:

$$d(p, q) := 2 \sinh^{-1}\left(\sqrt{\frac{2\|p - q\|^2}{c(1 - c\|p\|^2)(1 - c\|q\|^2)}}\right).$$

Refer to Figure 2(a) to visualize the geodesics in the Poincareé ball.

Thanks to the Killing-Hopf theorem Lang (1995), all hyperbolic space models with the same curvature and dimension are isometrically equivalent, indicating the distance between any pair of points is preserved after the transformation from one space to another. Thus, we can safely develop our methods on the Poincaré Ball without loss of generality.

### 4.3 Gyrovector Space

The traditional algebraic operations in Euclidean space are not directly applicable in the hyperbolic space. To perform algebraic operations like vector addition and scalar multiplication in hyperbolic space, we use the framework of *gyrovector spaces*, developed by Abraham A. Ungar Ungar (2022). These generalize vector spaces to hyperbolic settings and rely on non-associative operations (gyrogroups) instead of traditional vector addition.

The two primary operations we use are:

1. **Möbius Addition:** For $u, v \in \mathbb{D}^n$, the addition is defined as:
$$u \oplus_c v := \frac{(1 + 2c\langle u, v \rangle + c\|v\|^2)u + (1 - c\|u\|^2)v}{1 + 2c\langle u, v \rangle + c^2\|u\|^2\|v\|^2},$$
   where $c = -k$ is the curvature parameter. For subtraction, just substitute $v$ with $-v$.

2. **Möbius Scalar Multiplication:** For $r \in \mathbb{R}$ and $u \in \mathbb{D}^n$, we define:
$$r \otimes_c u := \frac{1}{\sqrt{c}} \tanh\left(r \tanh^{-1}(\sqrt{c}\|u\|)\right) \frac{u}{\|u\|}.$$

Refer to Figure 2(b) to comprehend both the functions pictorially. These operations form the backbone of many hyperbolic neural network formulations and allow us to compute metrics like the Davies-Bouldin Index or Calinski-Harabasz Score in a geometry-aware manner.

### 4.4 Exponential and Logarithmic Maps

These are two key functions for enabling transition between the hyperbolic manifold and its tangent (Euclidean) space. These functions are essential for optimization, embedding, and distance-based tasks.

Let $x \in \mathbb{D}^n$ and $v \in T_x(\mathbb{D}^n)$ be a tangent vector. Then:

The **Exponential Map** projects $v$ to a point on the manifold along a geodesic starting from $x$:
$$\exp_x^c(v) := x \oplus_c \left(\tanh\left(\sqrt{c}\frac{\lambda_x^c\|v\|}{2}\right) \frac{v}{\sqrt{c}\|v\|}\right),$$

The **Logarithmic Map** performs the reverse operation—mapping a point $y$ on the manifold back to the tangent space at $x$:
$$\log_x^c(y) := \frac{2}{\sqrt{c}\lambda_x^c} \tanh^{-1}\left(\sqrt{c}\|-x \oplus_c y\|\right) \frac{-x \oplus_c y}{\|-x \oplus_c y\|},$$
where the *conformal factor* is defined as $\lambda_x^c := \frac{2}{1-c\|x\|^2}$.

## 5 Proposed Method

In this section, we will unravel the design strategy of expansive Hyperbolic Deep Convolutional Neural Networks (eHDCNN). Let us first define the hyperbolic convolution operation on Poincaré Ball. Assume two functions $f$ and $g$ from $\mathbb{R}^n \to \mathbb{R}$, we define the convolution between $f$ and $g$ as:
$$f \star g(x) := \int_{\mathbb{R}^n} f(z)g(x - z)dz.$$

Analogously, we define hyperbolic convolution using logarithmic and exponential maps.

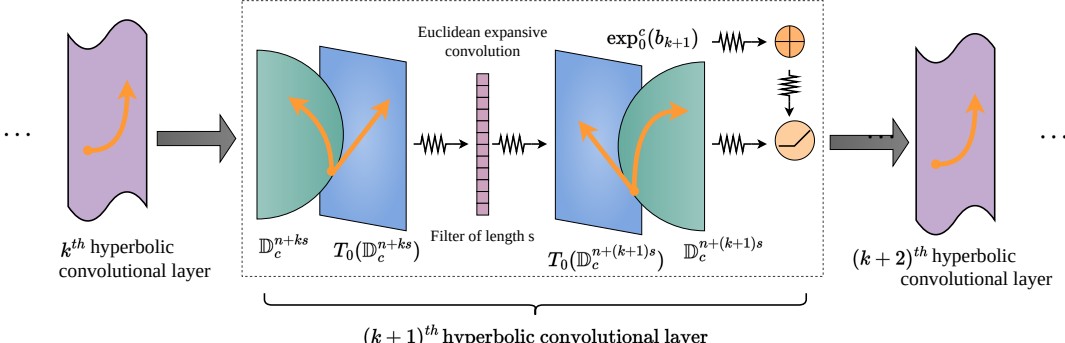

Figure 3: The complete workflow of expansive hyperbolic 1-d convolutional layer on Poincarê Ball is presented. (best view in digital format)

**Definition 1** (Hyperbolic Convolution (Continuous Version)). *For $x \in \mathbb{D}_c^n$, we define the convolutions of two real-valued functions $f, g$ on $\mathbb{R}^n$ as a map from $: \mathbb{D}_c^n \to \mathbb{D}_c^1$ as*

$$f \star g(x) := \exp_0^c \left[ \int_{\mathbb{D}_c^n} f(\log_0^c(z)) g(\log_0^c(-z \oplus_c x)) \lambda(z) \right], \tag{3}$$

*where $\lambda(z) := \frac{dz}{1-c\|z\|^2}$.*

**Remark 1.** *Note that for two real-valued functions $f, g$, their hyperbolic convolution is a map $h : \mathbb{D}_c^n \to \mathbb{D}_c^1$. We want to keep the range of the output function of the convolution in $\mathbb{D}_c^1$, since in the deep convolutional setup we will again convolute the output with some other filters.*

**Definition 2** (Hyperbolic Expansive and Contractive Convolution (discrete Version) ). *Let $\boldsymbol{w} := \{w_j\}_{j=-\infty}^{\infty}$ be an infinite dimensional vector whose elements are in $\mathbb{R}$ with finitely many non-zero entries in $\boldsymbol{w}$. Explicitly we assume $w_j \neq 0$ for $0 \leq i \leq s$. Among two widely used types of 1-D convolutions in $\mathbb{R}^n$, we talk about only the expansive and contractive type convolutions. Let $\boldsymbol{v} = \{v_1, ..., v_n\} \in \mathbb{D}_c^n$. We define the Hyperbolic Expansive Convolution ($*_h$) and the Hyperbolic Contractive Convolution ($\star_h$) in the following way:*

*Let $\boldsymbol{v'} := \log_0^c(\boldsymbol{v}) = (v_1', v_2', ..., v_n') \in \mathbb{T}_0^c(\mathbb{D}_c^n) \subseteq \mathbb{T}_0^c(\mathbb{R}^n)$, i.e. $\boldsymbol{v'}$ is an element of the tangent bundle at $0$ of $\mathbb{D}_c^n$.*

1. **Hyperbolic Expansive Convolution:** $(\boldsymbol{w} * \boldsymbol{v'}) = \sum_{l=1}^{n} w_{j-l} v_l'$ *for $j = 1, 2, ..., n + s$. Therefore $(\boldsymbol{w} * \boldsymbol{v'}) \in \mathbb{R}^{n+s}$. We apply the $\exp$ map to put it back in $\mathbb{D}_c^{n+s}$. Finally, we define*

$$\boldsymbol{w} *_h \boldsymbol{v} := \exp_0^c(\boldsymbol{w} * \log_0^c(\boldsymbol{v})). \tag{4}$$

2. **Hyperbolic Contractive Convolution:** *The usual contractive convolution for $\boldsymbol{w}$ and $\boldsymbol{v'}$ is defined as, $\boldsymbol{w} \star \boldsymbol{v'} = \sum_{l=j-s}^{j} w_{j-l} v_l, j = s + 1, ..., n$. We define $\star_h$ between $\boldsymbol{w}$ and $\boldsymbol{v}$ as*

$$\boldsymbol{w} \star_h \boldsymbol{v} := \exp_0^c(\boldsymbol{w} \star \log_0^c(\boldsymbol{v})), \tag{5}$$

*which lies in $\mathbb{D}_c^{n-s}$.*

**Remark 2.** *Note that for $c = 0$, we will retrieve the usual sparse Toeplitz operators of dimensions $n \times (n+s)$ and $n \times (n-s)$ from both contractive and expansive cases. Also, for $c = 0$, we retrieve the conventional 1-d Euclidean expansive and contractive convolutions; subsequently, our proposed architecture is reduced to its Euclidean variant by setting the $c = 0$.*

Having reviewed all the necessary terminology, we are now ready to define the complete architecture of the Hyperbolic Deep Convolutional Neural Network (HDCNN).

**Definition 3** (Hyperbolic Deep Convolutional Neural Network (HDCNN))**.** *Let $L \in \mathbb{N}$ be the number of hidden layers in the network. For a given set of filters $\{\boldsymbol{w_k}\}_{k=1}^{L}$ and set of compatible bias vectors $\{\boldsymbol{b_k}\}_{k=1}^{L}$ and a vector $\boldsymbol{a_L} = \{a_1, a_2, ..., a_{n_L}\}$ [Note that these vectors all lie in Euclidean Spaces of Appropriate Dimensions]. Let $\sigma(t) := \max\{0, t\}$ be the ReLU, acting component-wise for the multidimensional operation. We also assume the dimension of the output layer is $n_L$ and $\boldsymbol{h}_L(x) := \{h^1(x), h^2(x), ..., h^{n_L}(x)\} \in \mathbb{D}_c^{n_L}$. The HDCNN is defined as:*

$$\boldsymbol{h}_k(x) = \sigma(\boldsymbol{w}_k \circ \boldsymbol{h}_{k-1}(x) \oplus_c \exp_0^c(\boldsymbol{b}_k)), \tag{6}$$

*where $\boldsymbol{h}_k(x)$ is the output from the $k-$th hidden layer for $k \in \{1, 2, ..., L-1\}$ and $h_k(x) \in \mathbb{D}_c^{n+ks}$, where $\circ$ can be either $*_h$ or $\star_h$ as defined in Definition 2, $\boldsymbol{h}_0(x) = x$ and the final output is given as:*

$$h_L(x) = \exp_0^c [\boldsymbol{a}_L \cdot \log_0^c(\boldsymbol{h}_L(x))]. \tag{7}$$

The transformation of a vector, lying as a geodesic on Poincaré Ball, is shown in Figure 3 through hyperbolic convolution performed in an intermediate layer. This transformation projects the vector as another geodesic in a higher dimensional Poincaré Ball to the subsequent layer.

**Remark 3.** *If we restrict our focus only to the Expansive case (eHDCNN), note that the dimension of the input to each hidden layer is getting bigger by $s$ units every time. More explicitly, if we have started with $x \in \mathbb{D}_c^n$, and $\boldsymbol{w}_1$ is the first filter of length $s$, then $\boldsymbol{h}_1(x) \in \mathbb{D}_c^{n+s}$, which is the input dimension of the second hidden layer. Iteratively, the input dimension of the $k-$th hidden layer is as same as the dimension of $\boldsymbol{h}_{k-1}$, which lies in $\mathbb{D}_c^{n+(k-1)s}$. Finally, when we reach the output layer, the output dimension will be $n + Ls$, i.e., $n_L = n + Ls$. To make the Mö bius addition and the Möbius multiplications compatible, we need to have $\boldsymbol{a}_L \in \mathbb{R}^{n+Ls}$. Also note that for $c = 0$, this architecture is reduced to the eDCNN architecture described in Lin et al. (2022b).*

Table 1: Comparison of theoretical guarantees and expressivity properties between Euclidean DCNNs and hyperbolic DCNNs.

| Theoretical Property | Euclidean eDCNN | eHDCNN | Euclidean cDCNN | cHDCNN |
|---|---|---|---|---|
| Universal Approximation | ✓(proved) | ✓(proved) | ✗(still open) | ✗(still open) |
| Universal Consistency | ✓(under mild assumptions) | ✓(under mild assumptions) | ✗(not guaranteed) | ✗(not guaranteed) |
| Sample Complexity (Risk) | ✓$\widetilde{\mathcal{O}}(m^{-1/2})$ | ✓$\widetilde{\mathcal{O}}(m^{-1/2})$ | ✗unknown | ✗unknown |
| Geometry-Awareness | ✗Euclidean only | ✓explicitly curvature-aware | ✗Euclidean only | ✓explicitly curvature-aware |
| Expressivity for Hierarchies | ✗Limited | ✓Superior (due to hyperbolic embeddings) | ✗Limited | ✓Superior (due to hyperbolic embeddings) |
| Robustness to Distortion | ✗No geometric control | ✓Controlled via $\log_0^c$ and $\exp_0^c$ maps | ✗No geometric control | ✓Controlled via $\log_0^c$ and $\exp_0^c$ maps |

**Remark 4.** *Although we have mathematically formalized the notion of hyperbolic contraction convolution, our theoretical analysis focuses exclusively on the expansive case. This choice is primarily motivated by the universal approximation theorem for deep ReLU networks Hanin (2019), which guarantees universal approximation only when the network contains at least one hidden layer of width no less than $d + 1$, where $d$ is the input dimension. While it is possible to overcome this limitation by appending a fully connected ReLU network before the cDNN, identifying the appropriate depth and width of such a network introduces additional complexity, restricting the cDNNs from holding the universal approximation properties. In contrast, the universal approximation property naturally holds for eDCNNs Zhou (2020a), provided that none of the hidden layers has a width smaller than $d$. Furthermore, Table 1 presents a concise theoretical comparison between 1-d Euclidean DCNNs and hyperbolic DCNNs.*

# 6 Theoretical Analyses

We will now provide the proof for universal consistency following the framework established in Lin et al. (2022b). While we will appropriately generalize the results to the hyperbolic setting, it is first necessary to define some statistical terminologies to comprehend the mechanism of eHDCNN.

We consider a dataset $\mathcal{D} = \{z_i\}_{i=1}^m = \{x_i, y_i\}_{i=1}^m$, where the samples are assumed to be independent and identically distributed according to a Borel probability measure $\rho$ on the space $\mathcal{Z} = \mathcal{X} \times \mathcal{Y}$. Here, $x_i \in \mathcal{X} \subseteq \mathbb{D}_c^n$ and $y_i \in \mathcal{Y} \subseteq \mathbb{D}_c^1$. We assume $\mathcal{X}$ is a compact set for this discussion. The goal is to learn a function $f_{\mathcal{D}} : \mathcal{X} \to \mathbb{R}^1$ that minimizes the following *Hyperbolic $L^2$ Population Risk (HPR)*:

$$\mathcal{E}(f) := \int_{\mathcal{Z}} (f(x) - \log_0^c(y))^2 d\rho. \tag{8}$$

**Remark 5.** *The reason behind taking the* $\log$ *of* $y \in \mathcal{Y}$ *is that, the logarithm function will project back* $y \in \mathcal{Y}$ *to* $\mathbb{T}_0(\mathbb{D}_c^1) \subseteq \mathbb{R}^1$. *Hence, taking the difference between two real numbers will make sense. Also, if* $c \to 0$, *we will return the usual* $L^2$ *error on the Euclidean Spaces.*

Lemmas 1 and 2 can be viewed as natural extensions of classical results to the hyperbolic setting. Nevertheless, for the sake of completeness, we state them here and provide detailed proofs in the Appendix.

**Lemma 1.** *The Hyperbolic Regression Function (HRF)* $f_\rho(x) := \int_{\mathcal{Y}} \log_0^c(y) d\rho(y|x)$, *defined by the means of conditional distribution* $\rho(\cdot|x)$ *of* $\rho$ *at* $x \in \mathcal{X}$ *minimizes the HGE.*

The next lemma will deduce what we aim to minimize.

**Lemma 2.** *For any* $f : \mathbb{D}_c^n \to \mathbb{R}^1$, *we have*

$$\mathcal{E}(f) - \mathcal{E}(f_\rho) = \|f - f\rho\|_{L_{\rho_{\mathcal{X}}}^2},$$

*where* $\rho_{\mathcal{X}}(x) := \int_{\mathcal{Y}} \rho(x, y) d\mathcal{Y}(y)$, *for each* $x \in \mathcal{X}$, *the marginal distribution of* $\rho$ *on* $\mathcal{X}$.

The estimator that minimizes the hyperbolic population risk is the estimator that minimizes the empirical error over the class of all functions expressed by our eHDCNN architecture. Hence, the corresponding estimator or the *Empirical Risk Minimizer (ERM)* is defined as:

$$f_{\mathcal{D},L,s} := \arg \min_{f \in \mathcal{H}_{L,s}} \mathcal{E}_{\mathcal{D}}(f), \tag{9}$$

where

$$\mathcal{E}_{\mathcal{D}}(f) := \frac{1}{m} \sum_{i=1}^m (f(x_i) - \log_0^c(y_i))^2$$

denotes the empirical risk (HERM) associated with the function $f$ and for the filters $\boldsymbol{w_k}$ for $k \in \{1, 2, ..., L\}$ of length $s_k = d + ks$ and

$$\mathcal{H}_{L,s} := \{h_L(x), \boldsymbol{w}_k, \boldsymbol{b}_k \in \mathbb{R}^{d+ks}, k = 1, 2, ..., L\}$$

is the set of all hyperbolic outputs produced by the eHDCNN defined by 7.

To establish consistency, we must demonstrate that as the sample size $m \to \infty$, the sequence of estimators converges to the true value. Formally, a sequence of estimators is *strongly universally consistent* if it converges almost surely to the underlying parameter. In regression, this implies that empirical error estimators—derived via empirical risk minimization—converge to the generalization error over the space of square-integrable functions (a Hilbert space) with respect to the conditional output distribution. In the hyperbolic setting, we formalize this notion as follows:

**Definition 4.** *A sequence of Hyperbolic Regression Estimators (HRE) $(\{f_m\}_{m=1}^\infty)$ built through ERM is said to be strongly universally consistent if it satisfies the condition:*

$$\lim_{m\to\infty} \mathcal{E}(f_m) - \mathcal{E}(f_\rho) = 0$$

*almost surely, for every Borel probability distribution $\lambda$ such that $\log_0^c(\mathcal{Y}) \in L^2(\lambda_{(\mathcal{Y}|x)})$.*

The main result we will be going to prove here will be the following Theorem, which will prove the strong universal consistency of eHDCNN when the Hyperbolic Empirical Risk is minimized. The following Theorem considers a sequence of eHDCNNs as the universal approximators of continuous functions, where the depth of the network has been taken as a sequence depending upon the sample size of our dataset.

**Theorem 1.** *Suppose the following conditions hold as $m \to \infty$:*

1. *$L = L_m \to \infty$,*

2. *$M = M_m \to \frac{1}{\sqrt{c}}$,*

3. *$m^{-\theta} M_m^2 \left[1 + \frac{1}{M_m\sqrt{c}} \tanh^{-1}(M_m\sqrt{c})\right]^2 \to 0$,*

4.

$$\frac{A\log(B)}{m^{1-2\theta}} \to 0, \tag{10}$$

*where*

$$A := \left(\frac{1}{\sqrt{c}} \tanh^{-1}(M_m\sqrt{c})\right)^4 L_m^2(L_m + d)\log(L_m)$$

$$B := \left(\frac{1}{\sqrt{c}} \tanh^{-1}(M_m\sqrt{c})\right) m.$$

*hold for $\theta \in (0, 1/2)$ and input filter length as $2 \le s \le d$. Then $\pi_{M_m} f_{D,L_m,s}$ is strongly universally consistent, where $\pi_M(l) := \min\{M, |l|\} \cdot sign(l)$ is the well-known truncation operator.*

**Remark 6.** ***Justification of the constraints in Theorem 1:*** *At first glance, the conditions stated in Theorem 1 may appear somewhat arbitrary; however, each can be rigorously justified as necessary. The first condition, $L_m \to \infty$ as $m \to \infty$, pertains to the network depth and is crucial for ensuring the universal approximation capability of the eHDCNN, the Euclidean analogue has been discussed in Lin et al. (2022b). The second condition allows the input data points to increasingly approach the boundary of the Poincaré Ball $\mathbb{D}_c^d$ (with radius $1/\sqrt{c}$) as their number grows, which is achieved by relaxing the truncation parameter $M_m$. The third condition ensures that the squared distance of the data points from the center of $\mathbb{D}_c^d$ grows slower than the number of data points Györfi et al. (2002). Finally, the fourth condition states that the growth in the number of data points dominates the combined effect of the pseudo-dimension ($A$) and the metric entropy $(\log(B))$. This condition is essential to preserve stable training dynamics and is rooted in principles of concentration inequalities.*

**Remark 7.** *If we put $\lim c \to 0$ in Theorem 1, we get back Theorem 1 in Lin et al. (2022b). Therefore, Theorem 1 is a more generalized version, which is reduced to its Euclidean version for curvature $0$.*

**Remark 8.** *When we intend to perform the convergence analysis of a series in mathematical analysis, we first consider the partial sum of the series up to a certain term (let's say up to the $k-$th term) and then try to observe the behavior of the series by letting $k \to \infty$. This idea generates the involvement of the truncation operator in Theorem 1. Note that instead of taking $M_m \to \infty$ [which is used in Lin et al. (2022b)], we have made $M_m \to \frac{1}{\sqrt{c}}$ (letting our samples lie close to the boundary of the Poincaré Ball, whose radius is $\frac{1}{\sqrt{c}}$). As $M_m \to \frac{1}{\sqrt{c}}$, $\tanh^{-1}(M_m\sqrt{c}) \to \infty$, so does $M_m\left(\frac{1}{M_m\sqrt{c}} \tanh^{-1}(M_m\sqrt{c})\right)$. It will ease our work for giving an upper bound on the covering number of $\mathcal{H}_{L,s}$ in terms of the truncation*

*limit. Our adoption of the truncation operator is motivated by the widespread application of this operator in proving the universal consistency of various learning algorithms Györfi et al. (2002), Lin et al. (2022b).*

*Apart from the truncation operator in Theorem 1, several constraints are involved which are crucial to guarantee universal consistency. The constraint on depth $L_m \to \infty$ appears naturally as it is necessary for the universal approximation used in Lemma 7. The growth of the truncation limit concerning sample size $m$ is given by $m^{-\theta} M_m^2 \left[1 + \frac{1}{M_m \sqrt{c}} \tanh^{-1}(M_m \sqrt{c})\right]^2 \to 0$ instead of $M_m^2 m^{-\theta} \to 0$ [given in Lin et al. (2022b)] to incorporate the growth restriction of sample error in term of two increasing univariate functions $h_1(M_m)h_2(m^{-1})$, where $h_1(x) = x^2 \left[1 + \frac{1}{x\sqrt{c}} \tanh^{-1}(x\sqrt{c})\right]^2$ and $h_2(x) = x^\theta, \theta > 0$. Finally, the constraint in equation 10 will ensure the absolute difference between the generalization error and empirical error goes to 0, by enforcing the condition that the combined growth effect of the pseudo-dimension and the metric entropy of the class $\mathcal{H}_{L,s}$ is outpaced by the number of input samples, which will be used to prove Lemma 6.*

**Remark 9.** *Theorem 1 only demonstrates the universal consistency of the eHDCNN architecture for one-dimensional convolution. The primary restriction comes from the infeasibility of the convolutional factorization that appeared in Zhou (2020a) [also described in Lin et al. (2022b)]. Since the analysis in the hyperbolic set-up also relies on the universal approximation for the conventional eDCNN, the question of universal consistency remains open for two or higher-dimensional eHDCNN structures.*

We now dive into proving Theorem 1. Our main ingredient will be a version of Concentration Inequality [Theorem 11.4,Györfi et al. (2002)] after suitably adjusting the upper bound of the metric entropy concerning pseudo-dimension [Lemma 4, Lin et al. (2022b)]. Although our approach is similar to Lin et al. (2022b) to some extent, we have been able to derive a stronger version of Lemma 6 in Lin et al. (2022b) as presented in the proof of Lemma 6 in this paper, showing that the truncated empirical error converges to the truncated generalization error much faster in the case of hyperbolic convolution compared to the traditional Euclidean one. This will be established once we present our experimental results in terms of different curvatures (curvature 0 denotes the experiment has been done using eDCNN).

To prove Theorem 1 we divide our works into three parts as demarcated in Lin et al. (2022b) and will develop the appropriate hyperbolic versions of the corresponding results. We begin with expanding the bounds on the covering number for the class of functions defined in 7. We first need several terminologies.

Let $\nu$ be a probability measure on $\mathcal{X} \in \mathbb{D}_c^n$. For a function $f : \mathcal{X} \to \mathbb{R}$, we set

$$\|f\|_{L^p(\nu)} := \left(\int_\mathcal{X} |f(x)|^p \nu(x) d\mathcal{X}(x)\right)^{1/p}.$$

Denote by $L^p(\nu)$ the set of all functions with $\|f\|_{L^p(\nu)} < \infty$. For $\mathcal{A} \subseteq L^p(\nu)$, we denote $\mathcal{N}(\epsilon, \mathcal{A}, \|\cdot\|_{L^p(\nu)})$ the covering number of $\mathcal{A}$ in $L^p(\nu)$, which is the least number of balls of radius $\epsilon$ needed to cover up $\mathcal{A}$ with respect to the $\|\cdot\|_{L^p(\nu)}$ metric. In particular we denote $\mathcal{N}_p(\epsilon, \mathcal{A}, x_1^m) := \mathcal{N}_p(\epsilon, \mathcal{A}, \|\cdot\|_{L^p(\nu_m)})$, where $\nu_m$ is the emperical measure for the dataset $x_1^m := \{x_1, x_2, ..., x_m\} \in \mathcal{X}^m$. Further we define $\mathcal{M}(\epsilon, \mathcal{A}, \|\cdot\|_{L^p(\nu)})$ to be the $\epsilon-$packing number of $\mathcal{A}$ with respect to the $\|\cdot\|_{L^p(\nu)}$ norm, which is the largest integer $N$ such that given any subset $\{g_1, g_2, ..., g_N\}$ of $\mathcal{A}$ satisfies $\|g_i - g_j\| \geq \epsilon$ for all $1 \leq i < j \leq N$.

Next, we will mention lemma 9.2 from Györfi et al. (2002), which expresses a relation involving inequalities among the covering and packing numbers.

**Lemma 3.** *Let $\mathcal{G}$ be a class of functions from $\mathcal{X} \to \mathbb{R}$ and $\nu$ be a probability measure on $\mathcal{X}$. For $p \geq 0$ and $\epsilon > 0$, we have*

$$\mathcal{M}(2\epsilon, \mathcal{G}, \|\cdot\|_{L^p(\nu)}) \leq \mathcal{N}(\epsilon, \mathcal{G}, \|\cdot\|_{L^p(\nu)}) \leq \mathcal{M}(\epsilon, \mathcal{G}, \|\cdot\|_{L^p(\nu)}).$$

*In particular,*

$$\mathcal{M}_p(2\epsilon, \mathcal{G}, x_1^m) \leq \mathcal{N}_p(\epsilon, \mathcal{G}, x_1^m) \leq \mathcal{M}_p(\epsilon, \mathcal{G}, x_1^m).$$

Next, we have to derive an estimate of the upper bound of the Packing number for the pseudo dimension.

Since the Lemma 2, 3, and 4 from Capacity Estimates in Appendix A of Lin et al. (2022b) are taken from results proved on general metric spaces, we will just state Lemma 4 from Lin et al. (2022b) without proof in the context of hyperbolic space, which we will use later.

**Lemma 4.** *For $0 < \epsilon \leq M$ and $c^*$ being an absolute constant, we have*

$$\log_2 \sup_{x_m^1 \in \mathcal{X}^m} \mathcal{N}_1\left(\epsilon, \pi_M \mathcal{H}_{L,s}, x_1^m\right) \leq c^* L^2 (Ls + d) \log(L(s+d)) \log \frac{M}{\epsilon}.$$

We define the hyperbolic version of the generalization error (HGE) as

$$\mathcal{E}_{\pi_M}(f) := \int_{\mathcal{Z}} \left(f(x) - \log_0^c(y_M)\right)^2 d\rho, \tag{11}$$

and the *Hyperbolic Empirical Error (HEE)* (truncated) as

$$\mathcal{E}_{\pi_M, D}(f) := \frac{1}{m} \sum_{i=1}^m \left(f(x_i) - \log_0^c(y_{i,M})\right)^2, \tag{12}$$

where $l_M := \min\{M, |l|\} \cdot sign(l)$, the well known truncation operator.

We now provide a convergence criterion for the HEE estimates to the HGE estimate. We will use a hyperbolic version of the concentration inequality as given in Lemma 5, Lin et al. (2022b).

A more generalized version of Theorem 11.4 Györfi et al. (2002) can be presented as follows:

**Lemma 5.** *We assume $|y| \leq B$ and $B \geq \frac{1}{\sqrt{c}}$. For a set of functions $\mathcal{F}$ from $f : \mathcal{X} \to \mathbb{R}$ satisfying $|f(x)| \leq B$ and for all $m \geq 1$, we have*

$$\mathbb{P}[\exists f \in \mathcal{F} : \mathcal{E}(f) - \mathcal{E}(f_\rho) - (\mathcal{E}_D(f) - \mathcal{E}_D(f_\rho)) \geq \epsilon(\alpha + \beta + \mathcal{E}(f) - \mathcal{E}(f_\rho))]$$

$$\leq 14 \sup_{x_1^m \in \mathcal{X}^m} \mathcal{N}_1\left(\frac{\beta\epsilon}{20B}, \mathcal{F}, x_1^m\right) \exp\left(-\frac{\epsilon^2(1-\epsilon)\alpha m}{214(1+\epsilon)B^4}\right),$$

*where $\alpha, \beta > 0$ and $\epsilon \in (0, 1/2)$.*

Based on Lemma 5, the following Lemma will lay out the convergence criterion of the Truncated HEE estimates.

**Lemma 6.** *When $m^{-\theta} M_m^2 \left[1 + \frac{1}{M_m \sqrt{c}} \tanh^{-1}(M_m \sqrt{c})\right]^2 \to 0$ and equation 10 holds for $\theta \in (0, 1/2)$, then we have*

$$\lim_{m \to \infty} \mathcal{E}_{\pi_{M_m}}(\pi_{M_m} f_{D,L,s}) - \mathcal{E}_{\pi_{M_m}, D}(\pi_{M_m} f_{D,L,s}) = 0$$

*holds almost surely.*

Lemma 6 indicates that the truncated version of the hyperbolic $L^2$ risk approaches towards the truncated version of the hyperbolic empirical risk as $m \to \infty$ under the mentioned regularity constraints. We will further need this Lemma to complete the proof of Theorem 1.

We are finally in a position to prove Theorem 1; we will give our final lemma, which will complete the proof for universal consistency.

**Lemma 7.** *Let $\Omega \subseteq \mathbb{D}_c^d$ be compact and $2 \leq s \leq d$. Then for any $f \in \mathcal{C}(\Omega)$, there exist a sequence of filters $\boldsymbol{w}$ and bias vectors $\boldsymbol{b}$ of appropriate dimensions and $f_L^{\boldsymbol{w}, \boldsymbol{b}} \in \mathcal{H}_{L,s}$ such that*

$$\lim_{L \to \infty} \|f - f_L^{\boldsymbol{w}, \boldsymbol{b}}\|_{\mathcal{C}(\Omega)} = 0.$$

**Lemma 8.** *Sample Complexity. The rate of convergence of the empirical error to the minimum error is $\mathcal{O}(m^{-1/2})$, i.e.,*

$$\mathcal{E}(f_m) - \mathcal{E}(f_\rho) \leq c \; \mathcal{O}(m^{-1/2}), \tag{13}$$

*for some constant $c > 0$.*

**Remark 10.** *We notice from the proof of Lemma 6 that the truncated HEE estimates converge much faster to the corresponding HGE than their Euclidean equivalents. This property gives the eHDCNN architecture an edge over the eDCNN for faster training, with many fewer training iterations needed. Roughly speaking, since each layer is taking input from a Poincaré Ball, which in turn expresses the complex representation of the data to the next layer, even before the information gets carried out to the next layer directly from the previous layer, the architecture is very quick to learn the internal representation of the data. This will be evident from our simulation results, showing the ascendancy of our architecture over its Euclidean version to achieve lower error rates much faster for certain regression problems.*

**Remark 11.** *While our analysis broadly builds on the techniques developed by Lin et al. (2022b), the results presented here constitute a substantial generalization of their work. Our work is based on the prior assumption that the dataset lies in $\mathbb{D}_c^n$, for some $n \in \mathbb{N}$. While the statistical terminologies are adapted from Euclidean learning theory, their realization in hyperbolic spaces is nontrivial due to manifold-specific tools operations, such as Möbius addition, Möbius scalar multiplication, and $\exp_0^c$, $\log_0^c$ maps. In particular, Theorem 1 in our work recovers Theorem 1 of Lin et al. (2022b) as a special case in the limiting scenario where $c \to 0$. Furthermore, Lemma 6 establishes a curvature-dependent rate of convergence of the empirical error to the population error (up to a truncation) in the hyperbolic setting, as compared to the Euclidean case. This represents a notable improvement over the prior results of Lin et al. (2022b), suggesting the potential for better generalization performance in spaces with higher negative curvature. Moving a step further, we explicitly provide an expression for the sample complexity of the class $\mathcal{H}_{L,s}$, which was absent in earlier works.*

## 6.1 Computational Complexity of eHDCNN

In this subsection, we derive the computational complexity of the function class $\mathcal{H}_{L,s}$. The output of the $k$-th layer in an eHDCNN network resides in the hyperbolic space $\mathbb{D}_c^{d+ks}$. Applying the logarithmic map $\log_0^c$ to this layer involves $\mathcal{O}(d + ks)$ operations. The subsequent expansive convolution with a filter of length $d + ks$ requires $\mathcal{O}((d + ks)(d + (k+1)s))$ operations. Mapping the result back to the hyperbolic space $\mathbb{D}_c^{d+(k+1)s}$ via the exponential map $\exp_0^c$ adds another $\mathcal{O}(d + (k+1)s)$ operations. Therefore, the total computational cost of the $k$-th hyperbolic convolution layer amounts to:

$$\mathcal{O}(d + ks + (d + ks)(d + (k+1)s) + d + (k+1)s).$$

Summing this expression over all layers $k = 0$ to $L - 1$ yields a total cost of:

$$\mathcal{O}(sL^3 + dsL^2 + d^2L).$$

Additionally, the final layer of the class $\mathcal{H}_{L,s}$ incurs an extra $\mathcal{O}(d + Ls + d + Ls + d + Ls) = \mathcal{O}(d + Ls)$ operations. Hence, for $m$ input samples in a $d$-dimensional space, the overall computational complexity of the class $\mathcal{H}_{L,s}$ is: $\mathcal{O}(m(sL^3 + dsL^2 + d^2L))$.

Although eDCNNs do not involve the hyperbolic maps $\exp_0^c$ and $\log_0^c$, the expansive convolution still incurs a computational complexity of $\mathcal{O}(m(sL^3 + dsL^2 + d^2L))$. This is because, unlike the linear cost of the $\exp_0^c$ and $\log_0^c$ operations with respect to the input dimension, the expansive convolution itself exhibits a quadratic dependence in its conventional form. Consequently, while the linear-time costs introduced by the hyperbolic maps accumulate additively, the dominant quadratic nature of the expansive convolution ensures that the overall computational complexity remains effectively unchanged when transitioning from the Euclidean to the hyperbolic setting over the long term.

## 7 Experiments & Results

We will demonstrate the efficacy of eHDCNN with varying curvatures ($c = 0$ represents the conventional 1-dimensional CNN) by conducting experiments on synthetic and real-world datasets. Our Python-based implementation is available at https://anonymous.4open.science/r/eHDCNN-C4E6/README.md.

### 7.1 Synthetic Datasets

We will construct two regression tasks based on the following functions,

$$f(x) = \frac{\sin(\|x\|_2)}{\|x\|_2}, \qquad g(x) = \frac{\sqrt{\|x\|_2}}{1 + \sqrt{\|x\|_2}}.$$

We used the regression model $y = h(x) + \epsilon$ (where $h$ can be either $f$ or $g$) to generate the training samples, where $\epsilon \sim \mathcal{N}(0, 0.01)$ and $x \sim \text{unif}([-1, 1]^{10})$. A fixed set of 800/200 samples for the train/test split is used for the experiment, except that the test data are taken without the Gaussian noise. We have used a filter size of length 8 and the number of layers 4. We have trained our model over 100 iterations for 800 training samples and recorded the mean RMSE. We repeat the experiments for six different sets of curvatures. Refer to Figures 4(a) and 4(b) for the detailed illustration.

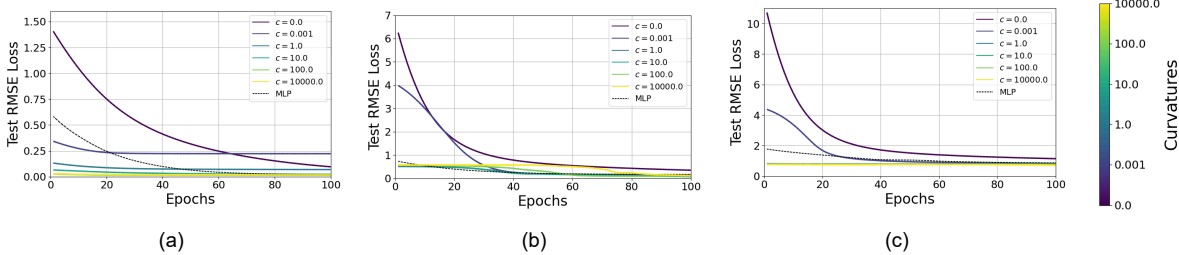

Figure 4: The performance analysis of eHDCNN with varying space curvatures (a) for $f(x)$ and (b) for $g(x)$, and (c) House price prediction is demonstrated. The Root Mean Square Error (RMSE) decreases faster with increasing curvature, justifying the utility of applying hyperbolic convolution. (best view in digital format)

The curves are evidence of the faster convergence of test RMSE loss during the entire training process, which validates the Remark 10. The loss curves are much steeper when the curvatures are more significant than zero compared to the same as the Euclidean counterpart. One point should be noted that the performance of eHDCNN started to deteriorate with the higher value of curvature. The phenomenon can be attributed to the contraction of Poincaré Ball with a very high curvature. Thus, the loss curves seem to be overlapping. Yet, the performance is commendable when eHDCNN is trained in the hyperbolic space of low curvature.

Table 2: The details of four real-world datasets are presented.

| Dataset | Superconductivity | Wave Energy Converters | House Price Prediction | WISDM |
|---|---|---|---|---|
| No of samples | 288000 | 21263 | 545 | 1073120 |
| No. of features | 81 | 81 | 12 | 3 |
| No. of classes | - | - | - | 6 |
| Target task | Regression | Regression | Regression | Classification |

### 7.2 Real-world Datasets

We considered four real-world datasets to showcase the effectiveness of eHDCNN. The details of the datasets and the hyperparameters are provided respectively in Table 2 and 3.

Table 3: The complete details of hyperparameters for four real-world datasets are presented to reproduce the results.

| Hyperparameters | Superconductivity | Wave Energy Converters | House Price Prediction | WISDM |
|---|---|---|---|---|
| No of layers | 4 | 4 | 4 | 4 |
| length of input filter | 8 | 8 | 8 | 9 |
| Noise | No | No | No | No |
| Learning Rate | 0.01 | 0.01 | 0.01 | 0.01 |
| Weight decay | 0.0005 | 0.0005 | 0.0005 | 0.0005 |
| Train/test split | 0.80 | 0.80 | 0.80 | 0.70 |
| No of samples | 288000 | 21263 | 545 | 1073120 |
| Input dimension | 81 | 81 | 12 | 240 |
| Batch Size | 128 | 128 | Full | 128 |
| Optimizer | Adam | Adam | Adam | Adam |

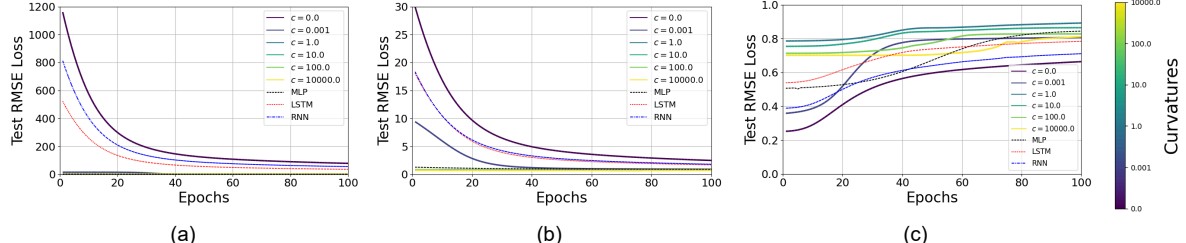

(a)  (b)  (c)

Figure 5: The performance analysis of eHDCNN with varying space curvatures (a) for Superconductivity, (b) for Wave Energy, and (c) test accuracy for WISDM is demonstrated. The Root Mean Square Error (RMSE) decreases faster for both (a) and (b) with increasing curvature compared to RNN, LSTM, MLP, Euclidean eDCNN ($c = 0$). Concurrently, test accuracy increases in (c), justifying the utility of employing hyperbolic convolution. For the RNNs and LSTMs, the number of layers remains fixed at 4. (best view in digital format)

.

### 7.2.1 Regression Task

We include 3 real-world regression datasets to demonstrate the performance of eHDCNN over the prevailing DCNN. We deploy the same eHDCNN architecture with 4 layers, and the length of the input filter is 8 for all three regression tasks. We split the entire dataset into 80% samples for training and the rest 20% samples for testing. We record the standardized test RMSE over the number of iterations during the training phase.

### House Price Prediction

We consider the widely available house price prediction dataset Wang & Zhao (2022) to solve the regression task. This dataset consists of 545 samples with 12 input features such as area, number of bedrooms, furnishing status, air conditioning, etc. At first, we standardize the entire data after numerically encoding its categorical column. We have trained our model using 4 layers and with an input filter length of 8. The test RMSE has been plotted against training iterations for six different curvatures in 4(c), where the curvature 0 means that the test RMSE has been plotted based on the eDCNN model.

### Superconductivity

As described in Hamidieh (2018), this dataset contains 21263 samples, each with 81 features like mean atomic mass, entropy atomic mass, mean atomic radius, entropy valence etc, along with the output feature as the critical temperature in the 82nd column. We split the dataset into 80 : 20 for our training and testing purposes. We will train our model with a mini-batch of size 128 in each training iteration. The test RMSE

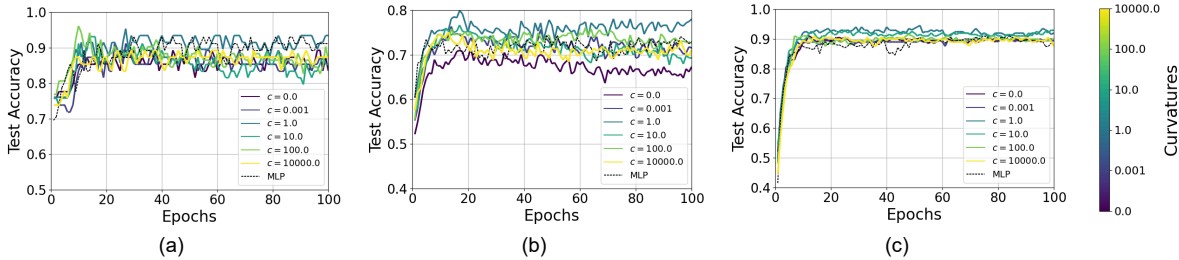

Figure 6: Prediction accuracy on three types of tasks by eHDCNNs at varying curvatures and MLP on the WordNet Dataset: (a) synset label classification task, (b) semantic relationship classification task, and (c) lexical category classification task

has been plotted against the number of training iterations for six different curvatures in 5 (a), where the curvature 0 indicates that the test RMSE has been taken based on the eDCNN model.

**Wave-Energy Converters**

As described in Mehdipour et al. (2024), this dataset contains 288000 samples, each with 81 features. This data set consists of positions and absorbed power outputs of wave energy converters (WECs) in four real wave scenarios from the southern coast of Australia (Sydney, Adelaide, Perth, and Tasmania). We split the dataset into 80 : 20 for our training and testing purposes. Similar to the Superconductivity dataset, we will train our model with a mini-batch of size 128 each time. For the test RMSE plot against the number of training epochs, we refer to 5 (b).

### 7.2.2 Classification Task

**WISDM**

We have applied eHDCNN on the WISDM, a well-adopted Human Activity Recognition (HAR) dataset Kwapisz et al. (2011). As it is described in Lin et al. (2022c), this dataset has six types of human activities such as cycling, jogging, sitting, standing, going upstairs and downstairs, with the corresponding accelerations along $x, y$, and $z$ axes at different timestamps and several user id ranging from 1 to 36. We have used the user IDs from 1 to 28 for training and the rest for testing. We have put 80 consecutive timestamps for each of the six classes together to make our input dimension $80 \times 3 = 240$. After this conversion, our training dataset has 10172 samples, and the test dataset has 3242 samples. Our experiment is carried out on a network with 4 layers with input filter length as 9. We have trained our model with a mini-batch of size 128 in each epoch. Although there is no inherent hierarchical structure in the WISDM dataset, it still exhibits a Gromov Hyperbolicity Index of only 0.1 when embedded as points on the Poincaré Ball of curvature 1, indicating a strong presence of latent hierarchies.

**WordNet**

We have applied eHDCNN on the WordNet dataset for three different classification tasks: *(i)* synset classification (node-level task), *(ii)* semantic relationship classification (edge level task), and *(iii)* part-of-speech (POS) tagging (word/text level task). The WordNet lexical database Miller (1995) organizes English words into sets of cognitive synonyms (synsets), each expressing a distinct concept. For all three tasks, we have utilized pre-trained GloVe embeddings of 100 dimensions to represent the input synsets or word pairs.

**Synset classification task** (6 (a)), Each input corresponds to the GloVe embedding of a synset's representative lemma in 100 dimensions, and the output is the corresponding lexical category (e.g., noun.animal, noun.food, verb.motion). The dataset consists of 59500 training samples and 14875 test samples, where the

synsets are drawn from various branches of the WordNet hierarchy. We perform stratified sampling to ensure balanced class distributions during training and evaluation.

**Semantic relationship classification task** (6 (b)), we consider synset pairs with annotated semantic relations such as *hypernym*, *hyponym*, *meronym*, and *antonym*. Each input is formed by concatenating the GloVe embeddings of the source and target synsets, yielding a 100-dimensional feature vector. The model is trained to predict the type of semantic relation between the pair. The training and test sets contain 67974 and 16994 examples, respectively, derived from well-curated edges of the WordNet graph.

**Lexical category classification/ part-of-speech (POS) tagging task** (6 (b)), the input consists of GloVe embeddings of surface word forms in 100 dimension, and the output is the corresponding coarse POS category, such as *noun*, *verb*, *adjective*, or *adverb*. We used a cleaned subset of WordNet where POS information is unambiguously tagged. The dataset is split into 39033 training and 9759 test samples, ensuring minimal lexical overlap between the splits to test generalization.

In all three tasks, we reshape the input embeddings into pseudo-spatial dimensions compatible with eHD-CNN, followed by a 4-layer convolutional hierarchy with input filter length 8. We train the models using mini-batches of size 128 for multiple epochs, optimizing MSE loss using the Adam optimizer. We have also included runtime comparisons for each task, presenting the execution time of the respective models on their corresponding datasets. In all cases, it is evident that the eHDCNN architecture yields improved accuracy at moderate curvature values. However, the decline in performance observed at higher curvature levels can be attributed to two primary factors: instability in training due to the use of Riemannian Stochastic Gradient Descent Mishne et al. (2023) or the Riemannian version of Adam Bécigneul & Ganea (2018), and the geometric property of the Poincaré ball, which tends to collapse towards a single point as $c \rightarrow \infty$, thereby reducing the model capabilities of representation learning.

### Results & Discussion

We run experiments on the House Price Prediction, Superconductivity, Wave-Energy Converters, WISDM, and WordNet datasets where plots can be seen respectively in Figures 4(c), 5(a), 5(b), 5(c), and 6. Test RMSE loss is the metric for the first three datasets, and test accuracy is the metric for the last one. The plots elucidate that the corresponding metric performs better when the curvature increases to a certain extent than the Euclidean variant. The better performance underscores the efficacy of hyperbolic architecture dominates over its Euclidean counterpart. Notably, our framework performs better than well-adopted recurrent architectures like RNN and LSTM applied to those 1-d tasks. One common point is that performance further degrades when the value of the curvature lies in a very high range. It occurs due to the shrinkage of the Poincaré Ball with a very high value of the curvature.

**Early-stage Saturation Problem** In Figures 4 and 5, the test RMSE loss almost saturates at the early stage of the training for the high value of space curvatures ($c \geq 100.0$). During the training process, if the embeddings are moved towards the boundary of the Poincaré ball, the vanishing gradient problem arises. Furthermore, for a Poincar/'e ball with higher curvature, the space is already contracted around its center. The training further pushes the embeddings to the boundary of the ball, resulting in lower gradients. Additionally, if the initialization parameter of the hyperbolic neural layers pushes the points to the boundary also experience gradient decay. Moreover, the problem is not observed when the space curvature is low.

### 7.3 Comparative Study of Complexity Analysis

We performed a comparative study on the time complexity of eHDCNN with various curvature settings and MLP. We measured the total training time for Euclidean DCNN (that is $c = 0$) with five distinct curvatures across 9 datasets. Refer to Figure 7 for the vivid illustration. The results suggest that our proposed eHDCNN might consume more time, which is compensated by the improved model performance. These results underscore the utility of 1-d expansive convolutions in the hyperbolic domain.

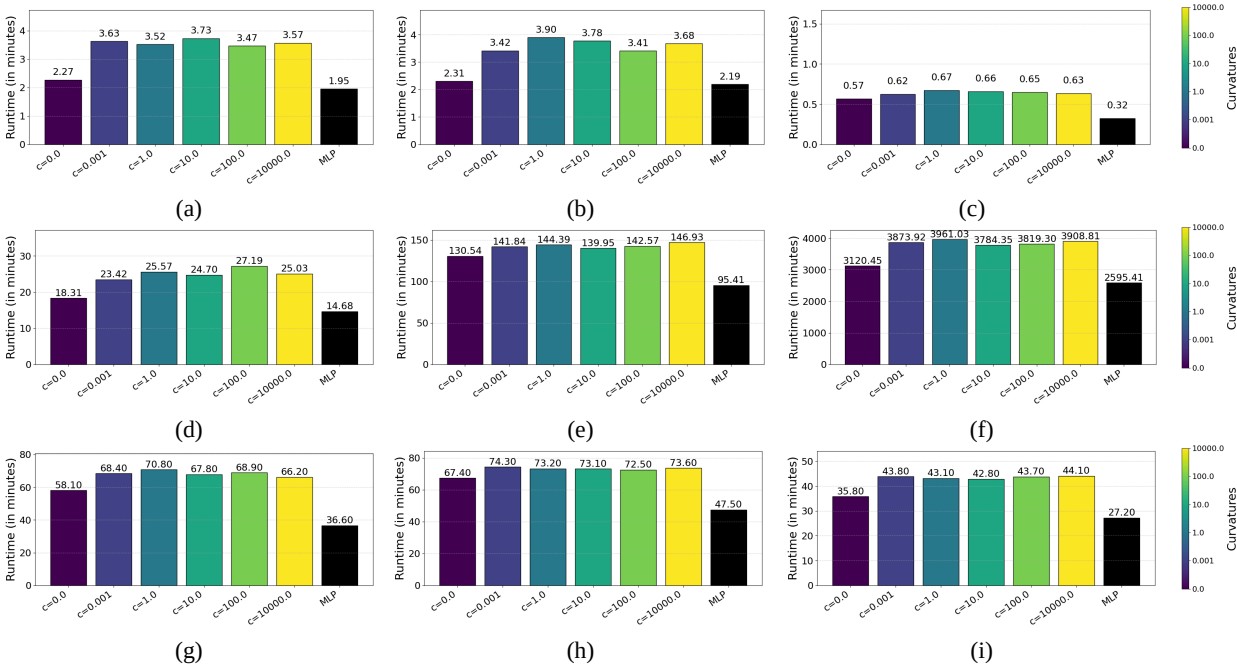

Figure 7: The comparative time complexity analysis among various curvatures of eHDCNN and MLP for the datasets (a) $f(x)$, (b) $g(x)$, (c) house price prediction, (d) Superconductivity, (e) Wave energy, (f) WISDM, (g) synset label classification task, (h) semantic relationship classification task, and (i) lexical category classification task are presented. Our approach requires some extra time compared to its Euclidean counterparts and MLP, but offers performance advantages.

## 8 Limitations

**Not applicable beyond 2 dimensions:** Our theoretical analysis is primarily restricted to 1- dimensional expansive hyperbolic convolutions. Although 2 and higher-dimensional convolutional architectures are commonly used in practice, convolutional factorization of the associated Toeplitz-type weight matrices becomes infeasible in higher dimensions Zhou (2020a). Consequently, the universal approximation properties of convolutional networks beyond one dimension remain an open problem to date Zhou (2020b). As a result, the theoretical results developed in this work cannot be directly extended to two- or higher-dimensional convolutional settings.

**Contractive CNNs are not universal approximators :** As noted in Remark 4, cDNNs are not universal approximators Hanin (2019), which restricts the applicability of universal consistency analysis using standard arguments. While universal approximation does hold for eDCNNs Zhou (2020a), it remains an open and intriguing research direction to investigate whether cDNNs can also approximate any continuous function on a compact domain.

## 9 Ablation Studies

We conduct an ablation study to study the effect of the filter length and number of hidden layers of the eHDCNN. The experiment is performed on Superconductivity. The filter length and number of layers are chosen respectively from the sets $s = \{6, 7, 8, 9\}$ and $L = \{3, 4, 5, 6\}$. We run experiments for each pair of $(s, L)$, and vary curvatures of the Poincaré Ball. The test RMSE curves are plotted and all results are presented in Figure 8. It can be observed that the test RMSE slowly decreases during the initial epochs of training of the eHDCNN. If we increase the number of layers or the length of the input filter, the respective error rates seem to be more stable and converge faster for the eHDCNN. This emphasizes the stability of

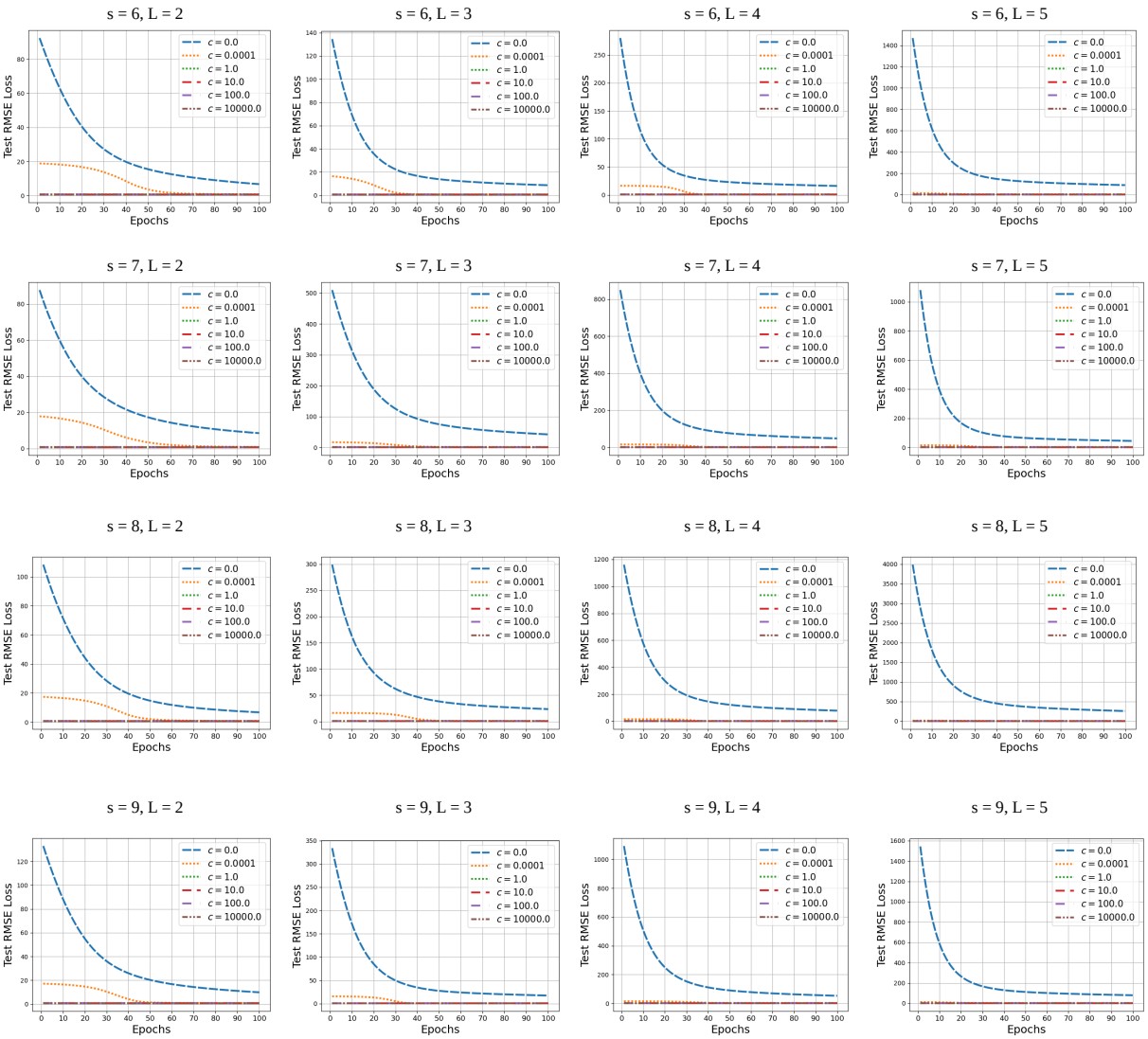

Figure 8: Various experiments were performed on the Superconductivity dataset by varying filter length and number of convolutional layers of the eHDCNN architecture.

our proposed architecture during training and is a clear indication of the fact that it requires a much lesser number of training iterations compared to the conventional eDCNN architecture for convergence.

## 10  Conclusion & Future Works

In this paper, we have identified the limitations of Euclidean spaces in providing meaningful information for training conventional DCNNs. We demonstrated the superiority of hyperbolic convolutions by treating the output of each layer as elements of the Poincaré Ball, projecting them onto the Tangent Space for expansive convolution, and then mapping them back to a higher-dimensional Poincaré Ball to capture complex hierarchical structures to the next layer. Our primary contribution is the proof of universal consistency by defining regression and error estimators in the hyperbolic space, drawing an analogy to Euclidean space. This is the first known result to explore the statistical consistency of architectures developed beyond the Euclidean domain. Furthermore, our simulation results validate our theoretical justification, showing why

eHDCNN is more adept at capturing complex representations, as noted in Remark 10. We anticipate that our findings will significantly accelerate the growth of deep learning spanning across the hyperbolic regime.

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

# A   Appendix: Proofs

We provide the detailed proofs and derivations of the Lemmas and Theorem presented in Section 6.

### Lemma 1

The Hyperbolic Regression Function (HRF) $f_\rho(x) := \int_\mathcal{Y} \log_0^c(y) d\rho(y|x)$, defined by the means of conditional distribution $\rho(\cdot|x)$ of $\rho$ at $x \in \mathcal{X}$ minimizes the *Hyperbolic Generalization Error (HGE)*.

*Proof.* The HGE can be written in terms of conditional expectation in the following way:

$$\mathcal{E}(f) = \int_\mathcal{Z} (f(x) - \log_0^c(y))^2 d\rho$$
$$= \mathbb{E}_{\mathcal{X},\mathcal{Y}}[f(\mathcal{X}) - \log_0^c(\mathcal{Y})]^2$$

Now for any function $g : \mathcal{X} \to \mathbb{R}^1$, we write

$$\mathcal{E}(g) = \mathbb{E}_\mathcal{X}\left[\mathbb{E}_{\mathcal{Y}|\mathcal{X}}\left[(g(\mathcal{X}) - \mathbb{E}[\log_0^c(\mathcal{Y})|\mathcal{X}] + \mathbb{E}[\log_0^c(\mathcal{Y})|\mathcal{X}] - \log_0^c(\mathcal{Y}))^2 |\mathcal{X}\right]\right]$$
$$= \mathbb{E}_\mathcal{X}\left[\mathbb{E}_{\mathcal{Y}|\mathcal{X}}\left[(g(\mathcal{X}) - \mathbb{E}[\log_0^c(\mathcal{Y})|\mathcal{X}])^2 |\mathcal{X}\right]\right] + \mathbb{E}_\mathcal{X}\left[\mathbb{E}_{\mathcal{Y}|\mathcal{X}}\left[(\mathbb{E}[\log_0^c(\mathcal{Y})|\mathcal{X}] - \log_0^c(\mathcal{Y})|\mathcal{X})^2 |\mathcal{X}\right]\right]$$
$$+ 2\mathbb{E}_\mathcal{X}\left[\mathbb{E}_{\mathcal{Y}|\mathcal{X}}\left[(g(\mathcal{X}) - \mathbb{E}[\log_0^c(\mathcal{Y})|\mathcal{X}])(\mathbb{E}[\log_0^c(\mathcal{Y})|\mathcal{X}] - \log_0^c(\mathcal{Y}))|\mathcal{X}\right]\right].$$

The cross term in the last expression is 0, since

$$\mathbb{E}_\mathcal{X}\left[\mathbb{E}_{\mathcal{Y}|\mathcal{X}}\left[(\mathbb{E}[\log_0^c(\mathcal{Y})|\mathcal{X}] - \log_0^c(\mathcal{Y}))\right]\right] = 0.$$

Therefore, the expression for HGE is reduced to

$$\mathcal{E}(g) = \mathbb{E}_\mathcal{X}\left[\mathbb{E}_{\mathcal{Y}|\mathcal{X}}\left[(g(\mathcal{X}) - \mathbb{E}[\log_0^c(\mathcal{Y})|\mathcal{X}])^2 |\mathcal{X}\right]\right] + \mathbb{E}_\mathcal{X}\left[\mathbb{E}_{\mathcal{Y}|\mathcal{X}}\left[(\mathbb{E}[\log_0^c(\mathcal{Y})|\mathcal{X}] - \log_0^c(\mathcal{Y})|\mathcal{X})^2 |\mathcal{X}\right]\right],$$

which attains minimum when $g(x) = \mathbb{E}\left[\log_0^c(\mathcal{Y})|x\right]$ for each $x \in \mathcal{X}$. Alternately, we write for each $x \in \mathcal{X}$

$$g(x) = \int_\mathcal{Y} \log_0^c(y) d\rho(y|x).$$

$\square$

### Lemma 2

For any $f : \mathbb{D}_c^n \to \mathbb{R}^1$, we have

$$\mathcal{E}(f) - \mathcal{E}(f_\rho) = \|f - f\rho\|_{L_{\rho_\mathcal{X}}^2},$$

where $\rho_\mathcal{X}(x) := \int_\mathcal{Y} \rho(x, y) d\mathcal{Y}(y)$, for each $x \in \mathcal{X}$, the marginal distribution of $\rho$ on $\mathcal{X}$.

*Proof.* Following the proof of Lemma 1, we can write

$$\mathcal{E}(f) - \mathcal{E}(f_\rho) = \mathbb{E}_\mathcal{X}\left[\mathbb{E}_{\mathcal{Y}|\mathcal{X}}\left[(f(x) - \mathbb{E}[\log_0^c(\mathcal{Y})|\mathcal{X}])^2\,|\mathcal{X}\right]\right]$$

$$= \mathbb{E}_{\mathcal{X},\mathcal{Y}}\big[\big[(f(x) - \mathbb{E}[\log_0^c(\mathcal{Y})|\mathcal{X}])^2\,|\mathcal{X}\big]$$

$$= \int_\mathcal{X}\int_\mathcal{Y}(f(x) - f_\rho(x))\rho(x,y)d\mathcal{X}(x)d\mathcal{Y}(y)$$

$$= \int_\mathcal{X}(f(x) - f_\rho(x))^2\int_\mathcal{Y}\rho(x,y)d\mathcal{Y}(y)d\mathcal{X}(x)$$

$$= \int_\mathcal{X}(f(x) - f_\rho(x))^2\rho_\mathcal{X}(x)d\mathcal{X}(x)$$

$$= \|f - f_\rho\|_{L^2_{\rho_\mathcal{X}}}.$$

□

## Lemma 3

Let $\mathcal{G}$ be a class of functions from $\mathcal{X} \to \mathbb{R}$ and $\nu$ be a probability measure on $\mathcal{X}$. For $p \geq 0$ and $\epsilon > 0$, we have

$$\mathcal{M}(2\epsilon, \mathcal{G}, \|\cdot\|_{L^p(\nu)}) \leq \mathcal{N}(\epsilon, \mathcal{G}, \|\cdot\|_{L^p(\nu)}) \leq \mathcal{M}(\epsilon, \mathcal{G}, \|\cdot\|_{L^p(\nu)}).$$

In particular,

$$\mathcal{M}_p(2\epsilon, \mathcal{G}, x_1^m) \leq \mathcal{N}_p(\epsilon, \mathcal{G}, x_1^m) \leq \mathcal{M}_p(\epsilon, \mathcal{G}, x_1^m).$$

*Proof.* The same proof mentioned in Lemma 9.2 Györfi et al. (2002), can be applied to any general metric space $M$ instead of $\mathbb{R}^d$. In particular $M$ can be $\mathcal{X}$. This shows the lemma is unaltered in the case of a compact subset in a hyperbolic space. □

## Lemma 5

We assume $|y| \leq B$ and $B \geq \frac{1}{\sqrt{c}}$. For a set of functions $\mathcal{F}$ from $f : \mathcal{X} \to \mathbb{R}$ satisfying $|f(x)| \leq B$ and for all $m \geq 1$, we have

$$\mathbb{P}[\exists f \in \mathcal{F} : \mathcal{E}(f) - \mathcal{E}(f_\rho) - (\mathcal{E}_D(f) - \mathcal{E}_D(f_\rho)) \geq \epsilon(\alpha + \beta + \mathcal{E}(f) - \mathcal{E}(f_\rho))]$$

$$\leq 14 \sup_{x_1^m \in \mathcal{X}^m} \mathcal{N}_1\left(\frac{\beta\epsilon}{20B}, \mathcal{F}, x_1^m\right) \exp\left(-\frac{\epsilon^2(1-\epsilon)\alpha m}{214(1+\epsilon)B^4}\right),$$

where $\alpha, \beta > 0$ and $\epsilon \in (0, 1/2)$.

*Proof.* The theorem 11.4 in Györfi et al. (2002) holds true for any set of functions $g : \mathbb{R}^d \to \mathbb{R}$. While in our constructions, the functions $f : \mathcal{X} \subseteq \mathbb{D}_c^d \to \mathbb{R}$. But we can still use the result of Theorem 11.4 by noting that $f \circ \log_0^c : \mathbb{R}^d \to \mathbb{R}$, i.e., consider by considering the class of functions $g = f \circ \log_0^c$ in Theorem 11.4 Györfi et al. (2002) instead of $f$ for all $f \in \mathcal{F}$. □

## Lemma 6

When $m^{-\theta}M_m^2\left[1 + \frac{1}{M_m\sqrt{c}}\tanh^{-1}(M_m\sqrt{c})\right]^2 \to 0$ and 10 holds for $\theta \in (0, 1/2)$, then we have

$$\lim_{m\to\infty}\mathcal{E}_{\pi_{M_m}}(\pi_{M_m}f_{D,L,s}) - \mathcal{E}_{\pi_{M_m},D}(\pi_{M_m}f_{D,L,s}) = 0$$

holds almost surely.

*Proof.* We have $|\pi_{M_m} f_{D,L,s}| \leq M_m$. And

$$|\log_0^c(y_{M_m})| = \left| \frac{1}{\sqrt{c}} \tanh^{-1}(\sqrt{c}\|y_{M_m}\|) \frac{y_{M_m}}{\|y_{M_m}\|} \right|$$
$$\leq \frac{1}{\sqrt{c}} \tanh^{-1}(\sqrt{c}M_m),$$

where the last inequality follows from the fact that $\tanh^{-1}$ is increasing on $(-1,1)$ and $\|y_{M_m}\| \leq M_m$, by truncation. Similarly, we have

$$|\log_0^c(y_{i,M_m})| \leq \frac{1}{\sqrt{c}} \tanh^{-1}(\sqrt{c}M_m).$$

Now from Equation 11, we write

$$\mathcal{E}_{\pi_{M_m}}(\pi_{M_m} f_{D,L,s}) = \int_{\mathcal{Z}} \left( f(x)_{M_m} - \log_0^c(y_{M_m})^2 \right) d\rho$$
$$\leq \int_{\mathcal{Z}} \left( M_m + \frac{1}{\sqrt{c}} \tanh^{-1}(\sqrt{c}M_m) \right)^2 d\rho$$
$$= M_m^2 \left[ 1 + \frac{1}{M_m\sqrt{c}} \tanh^{-1}(\sqrt{c}M_m) \right]^2. \tag{14}$$

Similarly, for the empirical version (Equation 12), we get

$$\mathcal{E}_{\pi_{M_m},D}(\pi_{M_m} f_{D,L,s}) \leq M_m^2 \left[ 1 + \frac{1}{M_m\sqrt{c}} \tanh^{-1}(\sqrt{c}M_m) \right]^2. \tag{15}$$

Combining inequalities 14 and 15 gives us

$$\left| \mathcal{E}_{\pi_{M_m}}(\pi_{M_m} f_{D,L,s}) - \mathcal{E}_{\pi_{M_m},D}(\pi_{M_m} f_{D,L,s}) \right| \leq 2M_m^2 \left[ 1 + \frac{1}{M_m\sqrt{c}} \tanh^{-1}(\sqrt{c}M_m) \right]^2. \tag{16}$$

Now putting $\alpha = \beta = 1$, in Lemma 5 and $\epsilon = m^{-\theta}$ we get that

$$\mathcal{E}_{\pi_{M_m}}(\pi_{M_m} f_{D,L,s}) - \mathcal{E}_{\pi_{M_m}}(f_\rho) - \left( \mathcal{E}_{\pi_{M_m},D}(\pi_{M_m} f_{D,L,s}) - \mathcal{E}_{\pi_{M_m},D}(f_\rho) \right)$$
$$\leq \epsilon \left[ \alpha + \beta + \mathcal{E}_{\pi_{M_m}}(\pi_{M_m} f_{D,L,s}) - \mathcal{E}_{\pi_{M_m}}(f_\rho) \right]$$
$$\leq m^{-\theta} \left[ 2 + 2M_m^2 \left( 1 + \frac{1}{M_m\sqrt{c}} \tanh^{-1}(\sqrt{c}M_m) \right) \right]^2 \tag{17}$$
$$= 2m^{-\theta} \left[ 1 + M_m^2 \left( 1 + \frac{1}{M_m\sqrt{c}} \tanh^{-1}(\sqrt{c}M_m) \right) \right]^2.$$

holds with probability at least

$$1 - 14 \sup_{x_1^m \in \mathcal{X}^m} \mathcal{N}_1 \left( \frac{1}{20\frac{1}{\sqrt{c}} \tanh^{-1}(M\sqrt{c})m^\theta}, \mathcal{F}, x_1^m \right) \exp \left( -\frac{m^{1-2\theta}}{428(1+\epsilon) \left( \frac{1}{\sqrt{c}} \tanh^{-1}(M\sqrt{c}) \right)^4} \right)$$

Here 17 follows from inequality 14. We assume $\mathcal{F} = \mathcal{H}_{L,s}$. Now we focus on providing a lower bound on

$$\sup_{x_1^m \in \mathcal{X}^m} \mathcal{N}_1 \left( \frac{1}{20\frac{1}{\sqrt{c}} \tanh^{-1}(M_m\sqrt{c})m^\theta}, \mathcal{F}, x_1^m \right) \exp \left( -\frac{m^{1-2\theta}}{428(1+\epsilon) \left( \frac{1}{\sqrt{c}} \tanh^{-1}(M_m\sqrt{c}) \right)^4} \right).$$

To do that, we will utilize Lemma 4 and the constraint 4 of Theorem 1. By putting $B = \frac{1}{\sqrt{c}}\tanh^{-1}(\sqrt{c}M_m)$ in Lemma 4, and using $\epsilon = m^{-\theta} \leq 1$, we write

$$\sup_{x_1^m \in \mathcal{X}^m} \mathcal{N}_1\left(\frac{1}{20\frac{1}{\sqrt{c}}\tanh^{-1}(\sqrt{c}M_m)m^\theta}, \mathcal{F}, x_1^m\right)\exp\left(-\frac{m^{1-2\theta}}{428\left(\frac{1}{\sqrt{c}}\tanh^{-1}(\sqrt{c}M_m)\right)^4}\right)$$

$$\leq \exp\left(c_2 \log\left(20\left(\frac{1}{\sqrt{c}}\tanh^{-1}(\sqrt{c}M_m)\right)m^\theta\right)L_m^2(d+sL_m)\log(L_m(s+d)) - c'\right), \qquad (18)$$

where $c' = \frac{m^{1-2\theta}}{428\left(\frac{1}{\sqrt{c}}\tanh^{-1}(\sqrt{c}M_m)\right)^4}$. Next, we focus on the argument of the exponential in 18. By writing $e = \log\left(20\left(\frac{1}{\sqrt{c}}\tanh^{-1}(\sqrt{c}M_m)\right)m^\theta\right)L_m^2(d+sL_m)\log(L_m(s+d))$ and $f = 428\left(\frac{1}{\sqrt{c}}\tanh^{-1}(\sqrt{c}M_m)\right)^4$, the argument inside the exponential in 18 becomes

$$c_2e - \frac{m^{1-2\theta}}{f} = -\left(m^{1-2\theta}/f\right)\left(1 - \frac{c_2ef}{m^{1-2\theta}}\right) = -\left(m^{1-2\theta}/f\right)\left(1 - \frac{c_3A\log(B)}{m^{1-2\theta}}\right), \qquad (19)$$

for some constant $c_3$, and $A$ and $B$ are as in Theorem 1. But the constraint 10 of Theorem 1 implies $\frac{A\log(B)}{m^{1-2\theta}} \to 0$ as $m \to \infty$. Hence, $c_2e - \frac{m^{1-2\theta}}{f} \to -\infty$ as $m \to \infty$, assuring inequality 17 holds almost surely [as the probability goes to 1 as $m \to \infty$].

Next, we invoke the Borel-Cantelli Lemma to show that $\mathcal{E}_{\pi_{M_m}}\left(\pi_{M_m}f_{D,L_m,s}\right) \xrightarrow{a.s.} \mathcal{E}_{\pi_{M_m},D}\left(\pi_{M_m}f_{D,L_m,s}\right)$. To this end, we compute

$$\lim_{r\to\infty}\sum_{m=1}^r \mathbb{P}\left[\mathcal{E}_{\pi_{M_m}}\left(\pi_{M_m}f_{D,L_m,s}\right) - \mathcal{E}_{\pi_{M_m},D}\left(\pi_{M_m}f_{D,L_m,s}\right) \geq 2m^{-\theta}\left[1 + M_m\left(1 + \frac{1}{\sqrt{c}M_m}\tanh^{-1}(\sqrt{c}M_m)\right)\right]^2\right]$$

$$\leq \lim_{r\to\infty}\sum_{m=1}^r \exp\left(c_2e - \frac{m^{1-2\theta}}{f}\right)$$

$$\leq \lim_{r\to\infty}\sum_{m=1}^r c_4 m \exp\left(-m^{1-2\theta}\right) \quad \text{[by Equation 19]}$$

$$\leq c_4\int_1^\infty xe^{-x^{1-2\theta}}dx$$

$$= c_4\Gamma\left(\frac{2}{1-2\theta}\right) < \infty,$$

for some constant $c_4 > 0$. Now, by the constraint (3) of Theorem 1,

$$2m^{-\theta}\left[1 + M_m\left(1 + \frac{1}{\sqrt{c}M_m}\tanh^{-1}(\sqrt{c}M_m)\right)\right]^2 \to 0$$

as $m \to \infty$. Therefore, the convergence $\mathcal{E}_{\pi_{M_m}}\left(\pi_{M_m}f_{D,L_m,s}\right) \to \mathcal{E}_{\pi_{M_m},D}\left(\pi_{M_m}f_{D,L_m,s}\right)$ holds almost surely as $m \to \infty$, completing the proof of Lemma 6.

$\square$

**Lemma 7**

Let $\Omega \subseteq \mathbb{D}_c^d$ be compact and $2 \leq s \leq d$. Then for any $f \in \mathcal{C}(\Omega)$, there exist a sequence of filters $\boldsymbol{w}$ and bias vectors $\boldsymbol{b}$ of appropriate dimensions and $f_L^{\boldsymbol{w},\boldsymbol{b}} \in \mathcal{H}_{L,s}$ such that

$$\lim_{L\to\infty}\|f - f_L^{\boldsymbol{w},\boldsymbol{b}}\|_{\mathcal{C}(\Omega)} = 0.$$

*Proof.* Define $g(y) := f(\exp_0^c(y))$ for $y \in \log_0^c(\Omega)$. Then, by Theorem 1 Zhou (2020a), we know that there exists $g_L^{\boldsymbol{w},\boldsymbol{b}}$ [where $g_L^{\boldsymbol{w},\boldsymbol{b}}$ lies in the free parameter space of the DCNN], such that

$$\lim_{L \to \infty} \|g - g_L^{\boldsymbol{w},\boldsymbol{b}}\|_{\mathcal{C}(\log_0^c(\Omega))} = 0.$$

We now define $f_L^{\boldsymbol{w},\boldsymbol{b}}(x) := g_L^{\boldsymbol{w},\boldsymbol{b}}(\log_0^c(x))$ for $x \in \mathbb{D}_c^d$. Now it is easy to verify that

$$\lim_{L \to \infty} \|f - f_L^{\boldsymbol{w},\boldsymbol{b}}\|_{\mathcal{C}(\Omega)} = \lim_{L \to \infty} \|g \circ \log_0^c - g_L^{\boldsymbol{w},\boldsymbol{b}} \circ \log_0^c\|_{\mathcal{C}(\log_0^c(\Omega))} = 0,$$

since the $\log_0^c$ [hence its inverse $\exp_0^c$] is global diffeomorphism from $\mathbb{D}_c^d \to \mathbb{R}^d$ [from $\mathbb{R}^d \to \mathbb{D}_c^d$]. $\qquad\square$

**Theorem 1**

Suppose $L = L_m \to \infty$, $M = M_m \to \frac{1}{\sqrt{c}}$, $m^{-\theta} M_m^2 \left[1 + \frac{1}{M_m\sqrt{c}} \tanh^{-1}(M_m\sqrt{c})\right]^2 \to 0$ [constrained truncation on the power of sample size] and

$$\frac{\left(\frac{1}{\sqrt{c}} \tanh^{-1}(M_m\sqrt{c})\right)^4 L_m^2 (L_m + d) \log(L_m)}{m^{1-2\theta}} \times \log\left(\left(\frac{1}{\sqrt{c}} \tanh^{-1}(M_m\sqrt{c})\right) m\right) \to 0,$$

(10) hold for $\theta \in (0, 1/2)$ and input filter length as $2 \le s \le d$. Then $\pi_{M_m} f_{D,L_m,s}$ is strongly universally consistent, where $\pi_M(l) := \min\{M, |l|\} \cdot sign(l)$ is the well-known truncation operator.

*Proof.* Since $M_m \to \frac{1}{\sqrt{c}}$, we have $M_m \times \left(\frac{1}{M_m\sqrt{c}} \tanh^{-1}(M_m\sqrt{c})\right) \to \infty$ as $m \to \infty$. We also have $\mathbb{E}[(\log_0^c(y))^2] < \infty$, i.e. $f_\rho \in L^2(\rho_{\mathcal{X}})$. By Lemma 7, we say that there exists a big enough $L_\epsilon$ so that $f_{L_\epsilon}^{w,b} \in \mathcal{H}_{L_\epsilon,s}$ with

$$\|f_\rho - f_{L_\epsilon}^{w,b}\|_{L^2(\rho_x x)}^2 \le \left[\limsup_{x \in \mathcal{X}} \|f_\rho(x) - f_{L_\epsilon}^{w,b}(x)\|\right]^2 = \left[\|f_\rho - f_{L_\epsilon}^{w,b}\|_{\mathcal{C}(\mathcal{X})}\right]^2 \le \epsilon,$$

where the second inequality follows from the fact that $\rho_{\mathcal{X}}$ being a Borel Probability measure on $\mathcal{X}$.

By the triangle inequality, we write

$$
\begin{aligned}
\mathcal{E}(\pi_{M_m}(f_{D,L,s})) - \mathcal{E}(f_\rho) \le & (\epsilon \mathcal{E}(\pi_{M_m}(f_{D,L,s})) - (1+\epsilon)\mathcal{E}(\pi_{M_m}(f_{D,L,s}))) \\
& + (1+\epsilon)\left(\mathcal{E}_{\pi_{M_m}}(\pi_{M_m}(f_{D,L,s}))) - \mathcal{E}_{\pi_{M_m},D}(\pi_{M_m}(f_{D,L,s}))\right) \\
& + (1+\epsilon)\left(\mathcal{E}_{\pi_{M_m},D}(\pi_{M_m}(f_{D,L,s})) - \mathcal{E}_{\pi_{M_m},D}(f_{D,L,s})\right) \\
& + (1+\epsilon)(\mathcal{E}_{\pi_{M_m},D}(f_{D,L,s})) - (1+\epsilon)^2(\mathcal{E}_D(f_{D,L,s})) \\
& + (1+\epsilon)^2\left(\mathcal{E}_D(f_{D,L,s}) - \mathcal{E}_D(f_{L_\epsilon}^{w,b})\right) + (1+\epsilon)^2\left(\mathcal{E}_D(f_{L_\epsilon}^{w,b}) - \mathcal{E}(f_{L_\epsilon}^{w,b})\right) \\
& + (1+\epsilon)^2\left(\mathcal{E}(f_{L_\epsilon}^{w,b}) - \mathcal{E}(f_\rho)\right) + ((1+\epsilon)^2 - 1)\mathcal{E}(f_\rho) \\
= & \sum_{i=1}^8 B_i.
\end{aligned}
$$

We will use an inequality, which we will require throughout the rest of the steps:

$$(s + t)^2 \le (1 + \epsilon)s^2 + (1 + 1/\epsilon)t^2 \tag{20}$$

for $s, t, \epsilon > 0$.

We will bound each of the $B_i$ to prove the universal consistency as done in Part 3 of Appendix A in Lin et al. (2022b).

We will start with $B_1$ as,

$$B_1 = \epsilon\mathcal{E}(\pi_{M_m}(f_{D,L,s})) - (1+\epsilon)\mathcal{E}(\pi_{M_m}(f_{D,L,s}))$$

$$= \int_{\mathcal{Z}} |\pi_{M_m}(f_{D,L,s}(x)) - (\log_0^c(y_{M_m})) + (\log_0^c(y_{M_m})) - (\log_0^c(y))|^2 d\rho$$

$$- (1+\epsilon)\int_{\mathcal{Z}} |\pi_{M_m}(f_{D,L,s}) - (\log_0^c(y_{M_m}))|^2 d\rho$$

$$\leq (1+(1/\epsilon))\int_{\mathcal{Z}} |\log_0^c(y) - \log_0^c(y_{M_m})|^2 d\rho.$$

But we have $M = M_m \to \frac{1}{\sqrt{c}}$ as $m \to \infty$. Since $\epsilon > 0$ is arbitrary, we get $B_1 \to 0$ as $m \to \infty$.

By Lemma 6 and the constraints in the statement of Theorem 1 we get

$$B_2 \to 0 \text{ as } m \to \infty.$$

By the definition of the truncation operator, we get,

$$B_3 = \frac{1}{m}\sum_{i=1}^{m} |\pi_{M_m}(f_{D,L,s}(x_i)) - (\log_0^c(y_{i,M_m}))|^2 - \frac{1}{m}\sum_{i=1}^{m} |f_{D,L,s}(x_i) - (\log_0^c(y_{i,M_m}))|^2 \leq 0.$$

By the Strong Law of Large Numbers and inequality 20 we have,

$$B_4 \leq (1+\epsilon)(1+1/\epsilon)\frac{1}{m}\sum_{i=1}^{m} |\log_0^c(y_i) - \log_0^c(y_{i,M_m})|^2$$

$$\to (1+\epsilon)(1+1/\epsilon)\int_{\ddagger} |\log_0^c(y) - \log_0^c(y_{M_m})|^2 d\rho$$

as $m \to \infty$ almost surely. By the fact that $M_m \to \frac{1}{\sqrt{c}}$ as $m \to \infty$, we get

$$B_4 \to 0.$$

Since $f_{D,L}$ is the estimator of Empirical Risk Minimizer, we obtain

$$B_5 = (1+\epsilon)^2 \left( \frac{1}{m}\sum_{i=1}^{m} |f_{D,L}(x_i) - \log_0^c(y_i)|^2 - \frac{1}{m}\sum_{i=1}^{m} |f_{L_\epsilon}^{w,b}(x_i) - \log_0^c(y_i)|^2 \right) \leq 0.$$

Again by the Strong Law of Large Numbers, we have

$$B_6 \to 0$$

almost surely.

For $B_7$ we have

$$B_7 = (1+\epsilon)^2\|f_{L_\epsilon} - f\rho\|_{L_{\rho_\mathcal{X}}^2}^2.$$

By Lemma 7, we get

$$B_7 \leq (1+\epsilon)^2\epsilon.$$

Also, we have

$$B_8 \leq ((1+\epsilon)^2 - 1)\int_{\mathcal{Z}} |f_\rho(x) - \log_0^c(y)|^2 d\rho = \epsilon(\epsilon+2)\int_{\mathcal{Z}} |f_\rho(x) - \log_0^c(y)|^2 d\rho.$$

Summing up all the terms from $B_1$ to $B_8$, we get

$$\limsup_{m\to\infty} \mathcal{E}(\pi_{M_m}(f_{D,L,s})) - \mathcal{E}(f_\rho) \leq (1+\epsilon)^2\epsilon + \epsilon(2+\epsilon)\int_{\mathcal{Z}} |f_\rho(x) - \log_0^c(y)|^2 d\rho \qquad (21)$$

holds almost surely. As $\epsilon > 0$ is arbitrary, we can write

$$\limsup_{m\to\infty} \mathcal{E}(\pi_M(f_{D,L,s})) - \mathcal{E}(f_\rho) = 0.$$

This completes the proof of the universal consistency of eHDCNN. $\square$

**Lemma 8 Sample Complexity:** The rate of convergence of the empirical error to the minimum error is $\mathcal{O}(m^{-1/2})$, i.e.,

$$\mathcal{E}(f_m) - \mathcal{E}(f_\rho) \leq c\ \mathcal{O}(m^{-1/2}), \qquad (22)$$

for some constant $c > 0$.

*Proof.* Note that, in the proof of Lemma 6, we substituted $\epsilon = m^{-\theta}, \theta \in (0, 1/2)$ into Lemma 4. Consequently, the convergence rate established in Lemma 6 was instrumental in proving Theorem 1, and in particular, Equation 21. Since the right-hand side of Equation 21 is linear in $\epsilon$, and the minimal approximation error $\int_{\mathcal{Z}} |f_\rho(x) - \log_0^c(y)|^2, d\rho$ is finite, we obtain:

$$\mathcal{E}(f_m) - \mathcal{E}(f_\rho) = \mathcal{O}(m^{-1/2}), \qquad (23)$$

thereby completing our claim regarding the sample complexity of the hypothesis class $\mathcal{H}_{L,s}$. $\square$

