# OpenReview forum: "On the Universal Statistical Consistency of Expansive Hyperbolic Deep Convolutional Neural Networks"
_TMLR — Rejected by TMLR_

### Review · Reviewer_tGRv · 2025-06-26

**Summary Of Contributions:**

This paper studies hyperbolic convolutions, as an extension of the widely used convolution in Euclidean spaces for neural networks. More precisely, the authors define a convolution operation in the Poincaré ball, a Riemannian manifold with constant negative curvature. The theoretical analysis is focused on the case of the expansive convolution, as opposed to the so-called contractive convolution. The main result of the paper regards the statistical consistency of such hyperbolic convolutional neural networks. After recalling some technical background on differential geometry and the Poincaré ball model, the authors present a construction of the hyperbolic convolution operations. The main theoretical result (theorem 1, statistical consistency) is then proved by extending existing arguments (already used in the Euclidean case) to the hyperbolic setting. In particular, it should be noted that the results recover exsiting ones in the limit of zero curvature. Finally, some numerical experiments show that hyperbolic network can converge faster than their Euclidean counterpart, even though instable results may appear when the absolute value of the curvature is too large.

**Audience:**

Yes

**Claims And Evidence:**

Yes

**Requested Changes:**

- Is the extension to 2 dimensional convolutions straightforward? If yes it might be worth discussing this fact, as it would have practical implications.
- The proof of Lemma 6 should be made clearer (see the points above) and corrected if necessary.
 - The same function g should be used both in Sections 2 and 7, also the choice of these functions should be motivated more.
 - Can you mention what is the optimiser used in the experiments?
 - Can you specify whether your results apply to contractive hyperbolic convolutions?
 - Can you provide further justifications of the assumptions of theorem 1? These assumptions do not look very natural at first.
 - Is it possible to obtain a non asymptotic result, ie, with an explicit convergence rate in $m$?

**Strengths And Weaknesses:**

**Strengths**

 - The statistical consistency result recover the Existing Euclidean results in the limit of zero curvature, hence, they are a strict generalization of these previous works.
 - The statistical consistency of hyperbolic convolutional networks doe snot seem to have been studied before and is an interesting topic.
 - The experiments support the idea that hyperbolic networks might be useful in practical applications.

**Weaknesses**

- It is mentioned that the focus of the paper is on the expansive convolution, but it is not clear why this choice is made and what part of the analysis is specific to the expansive case.
- You mention in your contributions that you define several statistical terminologies within the hyperbolic framework, but it seems to me that all these definitions are particular cases of the existing ones, so this might not be seen as a contribution (please correct me if I am wrong). Also, is the term generalization error standard for this quantity? To me Equation (8) should rather be called test error or population risk. Additionally, it looks like lemma 1 and 2 are direct consequences of classical results, it should be mentioned.
 - The proof of Lemma 6 is not very clear to me. First, the proof uses $M$ instead of $M_m$, so a dependence on $m$ is ignored (in particular the last line of the proof is weird). Second, at the end of the proof, why is exactly the strong law of large numbers mentioned? it is not clear from the proof how the high probability bound and the convergence result imply the almost sure convergence that is claimed. I think some details are missing in this proof.
- Lemma 5 is presented as a generalised version of an existing result, but no proof can be found in the appendix. More generally, all the proof seems to be a direct consequence of existing results (the main technical lemmas are taken from existing works). I think it should be made clearer where in the proofs does the hyperbolic setting change the techniques and what is specific to the hyperbolic case (why is it interesting to study this case in particular?).

**Minor remarks and typos**
 - There is a notation mismatch between Equation (9) and the equation just after.
 - $f_{D,L,s}$ is not formatted correctly in the first line of the proof of Lemma 6
 - proof of lemma 6: the word lemma is repeated twice page 19

---

> ### Comment · Reviewer_tGRv · 2025-07-21
> **About the proof of Lemma 6**
>
> Dear authors,
>
> I noticed that you have updated the proof of Lemma 6. I still have some questions about it.
>  - First, in the last equation of page 25, why mentioning $1-\delta$? Indeed, according to your derivations it should be $1$.
>  - Most importantly, I still don't understand how you can deduce the last statement of page 25 from Equations (17-19). Why is it possible to take the limit both inside and outside the probability. As I understand it you have prove statements of the type $\mathbb{P}(A_m \leq B_m) \geq \lambda_m \to 0$, with $B_m \to 0$ and you deduce that $\mathbb{P}(A_m \to 0) = \lim_{m\to \infty} \lambda_m$, why is it justified. For me there should be more details to ensure this is correct (maybe some kind of Borel-Cantelli argument?). Please correct me if I am mistaken.

---

> > ### Author Response · Authors · 2025-07-22
> > **Response to Reviewer tGRv**
> >
> > We thank the reviewer for insightful feedback. We have updated the manuscript, and modified portions are highlighted in blue.
> >
> > **R1 Proof of Lemma 6**: We appreciate the concern. We have further updated the proof of Lemma 6 with more mathematical rigor. The last argument requires a direct application of the *Strong Law of Large Numbers* and the *Borel-Cantelli Lemma*, and we have now modified the last part of the proof of Lemma 6.
> >
> > Hope this will enhance the clarity.

---

> > > ### Comment · Reviewer_tGRv · 2025-07-23
> > > **Thank you for your answer**
> > >
> > > Dear authors,
> > >
> > > Thank you for updating the proof according to my remarks, the Borel Cantelli argument looks correct to me.
> > > However, I don't understand why the strong law of large numbers is needed, for me it looks like the statement at the top of page 26 is not really correct, and not actually needed as there is the Borel Cantelli argument to prove it rigorously.

---

> > > > ### Author Response · Authors · 2025-07-23
> > > > **Response to Reviewer tGRv**
> > > >
> > > > We thank the reviewer for the suggestion once again. The *Strong Law of Large Numbers* is not required for the almost sure convergence in *Lemma 6*. We have modified the portion of the proof accordingly.
> > > >
> > > > We hope this will clarify the proof further.

---

### Review · Reviewer_FLMX · 2025-07-04

**Summary Of Contributions:**

This paper proposes expansive Hyperbolic Deep Convolutional Neural Networks (eHDCNN) based on the Poincaré Ball model and provides theoretical analysis of their universal statistical consistency. The authors extend traditional Euclidean DCNNs to hyperbolic space by defining hyperbolic convolution operations using logarithmic and exponential maps. The main theoretical contribution is proving universal consistency for 1-D expansive hyperbolic convolution, claiming this is the first such result for fully hyperbolic architectures. Experimental validation is provided on synthetic and real-world datasets, showing faster convergence compared to Euclidean counterparts.

**Audience:**

Yes

**Claims And Evidence:**

No

**Requested Changes:**

- Include a table to clearly show the theoretical results comparison between regular CNN and eHDCNN, including universal consistency, universal approximation, and related properties.

- Clarify the Motivating Example in Sec. 2 and Synthetic Datasets in Sec. 7.1:
    - Why choose f(x) = sin(‖x‖₂)/‖x‖₂ and g(x) = cos(‖x‖₂)/‖x‖₂ to demonstrate the effectiveness of hyperbolic expansive convolution? The authors mention Gromov Hyperbolicity values (δf = 0.45, δg = 0.13) but don't explain why these specific functions are representative of problems where hyperbolic geometry should excel.
- Clarify why the eHDCNN has very low test error at the 0 epoch without any training (shown in Fig. 3 and 4):
   - This is unusual and may suggest potential issues since networks should not achieve near-optimal performance before training begins.

- Clarify that does eHDCNN with c=0.0 recover to regular CNN as used in practice:
   - While the authors claim in Remark 2 that c=0 retrieves usual operators, it's unclear if this exactly matches standard CNN implementations in practice, which would affect the validity of the baseline comparison.


- Missing baselines for experiments comparison, such as regular MLP and 1D CNN:
   - The paper only compares against the c=0.0 case but lacks comparison with standard deep learning baselines like multilayer perceptrons or properly implemented 1D CNNs, making it difficult to assess the true performance gains.

**Strengths And Weaknesses:**

Strengths
- The paper provides rigorous proof of universal statistical consistency for hyperbolic convolutional neural networks, which is a significant theoretical advancement in extending statistical learning theory to non-Euclidean spaces.

Weaknesses
- Inconsistency between motivation and experimental validation:
   - In Section 1, the authors claim that hyperbolic NNs can better represent complex relationships and hierarchical structures in datasets, specifically mentioning that "Learning embeddings of hierarchical data in Euclidean spaces often falls short in capturing meaningful structural information.
   - However, experiments are conducted on relatively simple/small datasets (synthetic functions, house prices, superconductivity, wave energy, WISDM) that don't clearly demonstrate hierarchical structure. None of these datasets exhibit the tree-like or graph-like hierarchical properties that supposedly motivate the use of hyperbolic geometry.
   - The choice of datasets and experimental setup doesn't convincingly justify the need for hyperbolic geometry over standard Euclidean approaches.

- Missing computational complexity analysis:
    - The paper fails to discuss the computational overhead and practical tradeoffs of hyperbolic operations compared to regular CNNs. The log/exp maps and Möbius operations likely introduce significant computational costs that should be quantified and weighed against the claimed benefits.

---

> ### Author Response · Authors · 2025-07-18
> **Response to Reviwer FLMX**
>
> We thank the reviewer for providing valuable feedback and insightful criticisms of our work. We made several updates in the manuscript. The modified portions are highlighted in dark brown.
>
> **R1 Incostintency between motivation and experimental validation**: To establish the efficacy of eHDCNN, we conducted experiments on a familiar hierarchical dataset, WordNet [1], a large English lexical database. The dataset encompasses three distinct tasks like (1) synset classification task, (2) semantic relationship classification task, and (3) lexical category classification or part-of-speech (POS) tagging task. We applied our framework, eHDCNN, to solve those tasks, and the results are presented in Figure 5 of the updated manuscript. We achieved better performance in terms of test accuracy for each task compared to Euclidean eDCNN (where $c=0$) and Multi-layer perceptron (MLP). Furthermore, the better performance indicates the capability of our proposed hyperbolic convolution to effectively capture the hierarchical characteristics present in the dataset.
>
> [1] George A Miller. Wordnet: a lexical database for english. Communications of the ACM, 38(11):39–41, 1995.
>
> **R2 Computational complexity**: We derive the computational complexity of eHDCNN. We also compare the complexity with the conventional eDCNN. In both cases, we found the time complexity will be $\mathcal{O}(m(sL^3 + dsL^2 + d^2L))$ where $m=$no. of samples, $d=$feature dimension, $L=$no of convolutional layers, $s=$filter length. The operations like $\log^{c}_{0}$ and $\exp^{c}_{0}$ yields linear time with respect to input dimension. Thus, applying hyperbolic convolutions will not affect the overall time complexity. Additionally, we offer a comparative study in Figure 3 on the training time of 9 different datasets used in the work. We include eDCNN (Euclidean version) and MLP as our baselines, along with different curvature settings. The analysis reveals that our method takes some extra amount of training time compared to eDCNN and MLP. The impressive performances of eHDCNN underscore the significance of hyperbolic 1-d convolution. More details can be found in Section 6.1
>
> **R3 Table addition for comparative studies**: We have performed a comparative study of the theoretical guarantees and expressivity between the Euclidean DCNN and hyperbolic DCNN in Table 1. The deeper observation reveals that the hyperbolic variant possesses more important statistical properties compared to its Euclidean variant. This also underscores the utility of deriving the theoretical properties of the hyperbolic variants.
>
> **R4 Motivating examples**: The choice of the function $f$ and $g$ are heavily reliant on the metric $\delta$ Gromov Hyperboliciy (GH) [2]. A low GH value denotes the ‘tree-likeliness’ of the underlying point set. In our case, both functions achieved a lower hyperbolicity value, indicating that the generated point sets contain hierarchical structures. Therefore, we considered these two functions as the potential candidates to validate our approach.
>
> [2] Jussi Väisälä. Gromov hyperbolic spaces. Expositiones Mathematicae, 23(3):187–231, 2005.
>
> **R5 Low test error in the early stage of training**: The early stage saturation problem persists when the space curvatures are higher, mostly greater than $100$. The problem can be attributed if the embeddings are pushed towards the boundary of the Poincaré ball, which causes vanishing gradient. Even with poor initialization causes the points are pushed in such regions where they experience gradient decay. The ball with higher curvatures is contracted around the center, which causes the embeddings to be closer to the boundary, resulting in the vanishing gradient. More details can be found in the Subsection ‘Results and Discussion’.
>
> **R6 Recovery of Euclidean eDCNN from the eHDCNN**: In the equations 4 and 5, putting $c=0$, the original Toeplitz operators of dimensions $n\times (n+s)$ and $n\times (n-s)$ are recovered after performing some algebraic manipulations. Thus, for $c=0$, we recover the Euclidean expansive and contractive convolutions. We summarize the facts in *Remark 2* of the updated manuscript.
>
> **R7 Missing baselines**: We further executed experiments on the MLP for each of the datasets mentioned in the manuscript and compared them with our existing results. Notably, eHDCNN with $c=0$ eventually reduces to the $1$-d CNN architecture that we have previously compared in the entire experiments. Now, we have successfully included all the baselines as suggested by the reviewer.

---

### Review · Reviewer_BJKC · 2025-07-15

**Summary Of Contributions:**

This paper defines a (presumably novel) hyperbolic convolutional layer. It provides a theoretical proof of the universal consistency of this operator (but I am unqualified to evaluate these proofs). They provide experiments on small networks on small-scale datasets that show that their hyperbolic CNN outperforms the euclidean CNN.

**Audience:**

Yes

**Claims And Evidence:**

Yes

**Requested Changes:**

- Fix typos and axis alignment in Fig 1
- Potentially more experimental baselines, but perhaps I am just out of the loop on this particular subfield -- it's hard for me to contextualize whether this is an important contribution

As it stands, I can't recommend acceptance based on presentation alone. I'll defer evaluation of the technical details to other reviewers..  I don't think the experiments are compelling.

**Strengths And Weaknesses:**

### Strengths

- The proposed convolution operation seems to be theoretically principled
- Experimental results show that the hyperbolic models outperform the Euclidean baselines in terms of RMSE or test accuracy
- Many experimental settings across different numbers of layers and filter sizes (though I find the graphs to be a bit overwhelming in terms of number...)
- Requires fewer iterations to converge than the Euclidean networks, but unclear if actually faster for inference/training time?

### Weaknesses

- Presentation is overall quite poor:
   - the amount of background was for me insufficient to evaluate the paper as a non-expert in this specialization, but large enough to almost read like a textbook in places.
    - Figures are kind of hard to read with various untied y-axes; for example, in Figure 1 it's hard to compare the lhs and rhs because the y axes are different scales
    - Numerous weird typos seemingly from a search/replace with disc->ball -- this simply doesn't inspire confidence and the carelessness made me skeptical of the paper overall
- Computer vision was a motivating example, unclear why this is only for small 1d tasks
- Comparison to other time series modeling techniques would be good

---

> ### Author Response · Authors · 2025-07-26
> **Response to Reviewer BJKC**
>
> We thank the reviewer for the valuable feedback and insightful criticisms of our work. We have made several updates to the manuscript. The modified portions are highlighted in dark orange.
>
> **R1 Quality of Presentation**: We have updated the preliminaries pertaining to Riemannian manifolds and hyperbolic geometry to easily comprehend the rest of our theoretical analyses. Every terminology, key terms, and functions are explained elaborately. We further represent the fundamental concepts pictorially in Figure 2 to improve readability. Please refer to section *Preliminaries* for more details.
>
> We have updated the figures to enhance the quality.
>
> We have updated the $y$-axes of both plots in Figure 1 to the identical scales for fair comparison.
>
> We have rectified all linguistic errors to enhance the clarity of our work.
>
> **R2 Difference from 2d Convlution**: Our entire theoretical analysis encompasses hyperbolic $1$-d convolution. Although higher than $1$-dimensional convolutional architectures are commonly used in the domain of Computer Vision. In this context, the convolutional factorization of the associated Toeplitz-type weight matrices becomes infeasible in higher dimensions [1]. Consequently, the universal approximation properties of convolutional networks beyond one dimension remain an open problem in the literature [2]. As a result, the theoretical results developed in our work cannot be directly extended to higher-dimensional convolutional settings.
>
> [1] Ding-Xuan Zhou. Universality of deep convolutional neural networks. Applied and computational harmonic analysis, 48(2):787–794, 2020a.
>
> [2] Ding-Xuan Zhou. Theory of deep convolutional neural networks: Downsampling. Neural Networks, 124: 319–327, 2020b.
>
> **R3 Comparison to Time Series Modeling**: We appreciate the suggestion and conducted experiments on Superconductivity, Wave Energy, and WISDM, including two well-adopted time series models, RNN and LSTM. The results are presented in Figure 5. The test RMSE loss of RNN and LSTM converges slowly compared to eHDCNN with varying curvatures. Such patterns underscore the effectiveness of eHDCNN over the conventional recurrent architectures for datasets that probably contain hierarchical structures.
>
> **R4: Validation through experiments**: We conducted a diverse set of experiments to validate the efficacy of eHDCNN. We have considered four standard datasets: Superconductivity, Wave Energy Converters, House Price Prediction, and WISDM, along with two synthetic benchmarks as motivating examples. Additionally, we have experimented on a familiar hierarchical dataset, WordNet [3], a large English lexical database. The dataset encompasses three distinct tasks like (1) synset classification task, (2) semantic relationship classification task, and (3) lexical category classification or part-of-speech (POS) tagging task.  We compared with Euclidean DCNN, MLP, RNN, LSTM, and with various curvature settings. The contextual discussion of the results is provided in the manuscript. Furthermore, we performed an extensive computational complexity analysis with several baselines to establish the effectiveness of our approach.
>
> [3] George A Miller. Wordnet: a lexical database for english. Communications of the ACM, 38(11):39–41, 1995.

---

### Decision · Action_Editor_rLNK · 2025-09-04

**Recommendation:** Reject

**Audience:**

Yes

**Audience Explanation:**

The paper extends classical CNNs to hyperbolic space, a relevant research topic within the machine learning community.

**Claims And Evidence:**

No

**Claims Explanation:**

**Summary**

The paper extends 1D DCNNs to hyperbolic space, proving universal consistency for 1D expansive hyperbolic convolution and providing empirical results on convergence. During rebuttal, some errors in the proofs were corrected. The final reviewer recommendations are split, with two leaning toward acceptance and one leaning toward rejection.

***

**Recommendation**

I recommend rejecting the paper at this time. I am not fully convinced of the paper's correctness or the significance of its results due to several outstanding issues.

***

**Key Concerns**

* A reviewer raised an important concern about the correctness of Lemma 6. The fact that resolving this issue required multiple rounds of rebuttal undermines the paper's theoretical validity, which is especially critical for a theoretical paper. Quote " ... the clear mistakes that were found make me doubt the validity of the paper."

* There is a reviewer consensus that the experimental results are underwhelming. Quote "...the practical impact may currently appear modest based on the limited experimental results on small datasets. "

***

**Additional Points for Improvement**

* **Presentation Quality:** The paper requires thorough proofreading to fix typos and notational inconsistencies (e.g., use of bolding, sub-indices in the first pages).
* **Experimental Validity:** The result that randomly initialized HDCNNs outperform well-trained DCNNs (Figures 4/8) and the initial loss is huge is highly unusual and indicates potential issues (incorrect initialization, normalization, etc) with the experimental design.
* **Missing Discussion of the role of Channels:** A crucial architectural detail, the **number of channels (network width)**, isn't discussed in both the theoretical and experimental sections. This concept (# of channels) is critical of modern neural networks, for both expressivity and generalization.
* **Baselines:** The experiments rely on author-created baselines rather than common, well-accepted benchmarks, which weakens the credibility of the results.

In light of these issues and the fact that reviewers feel unconfident about the theoretical and/or empirical results, I am unable to recommend this paper for publication in TMLR.